# STRUCTURED INVERSE-FREE NATURAL GRADIENT: MEMORY-EFFICIENT & NUMERICALLY-STABLE KFAC FOR LARGE NEURAL NETS

## ABSTRACT

Second-order methods for deep learning—such as KFAC—can be useful for neural network training. However, they are often memory-inefficient and numerically unstable for low-precision training since their preconditioning Kronecker factors are dense, and require high-precision matrix inversion or decomposition. Thus, such methods are not widely used for training large neural networks such as transformer-based models. We address these two issues by (i) formulating an *inverse-free* update of KFAC and (ii) imposing *structures* in each of the Kronecker factors, resulting in a method we term *structured inverse-free natural gradient descent (SINGD)*. On large transformer- and convolution-based models, we show that, in contrast to KFAC, SINGD is memory efficient and numerically robust.

## 1 INTRODUCTION

The continuing success of deep learning (DL) is—to a large extent—powered by scaling up computational power (Thompson et al., 2020) to increase the number of neural network (NN) parameters that can be trained. Contemporary natural language processing (Radford et al., 2019; Brown et al., 2020; Touvron et al., 2023) and computer vision (Dehghani et al., 2023) models often consist of many billions of parameters, and will likely grow further in the future. To compensate for increasingly higher computational demands of training more parameters, many training pipelines use lower precision data types (Micikevicius et al., 2018) and memory-efficient first-order optimizers like SGD (Robbins & Monro, 1951) or Adam(W) (Kingma & Ba, 2015; Loshchilov & Hutter, 2019).

Second-order methods, like natural gradient descent (NGD, Amari, 1998), leverage curvature information which has many applications in deep learning: It is useful for improving training dynamics (Martens & Grosse, 2015; Osawa et al., 2023), pruning (Wang et al., 2019), understanding the influence of training examples (Bae et al., 2022), and uncertainty estimation (Zhang et al., 2018; Immer et al., 2021; Daxberger et al., 2021). One major obstacle why those methods are rarely used in deep learning is their higher memory consumption and iteration cost.

The perhaps most common concept to scale second-order methods for DL is Kronecker-factored approximate curvature (KFAC, Heskes, 2000; Martens & Grosse, 2015) which approximates the Fisher's block diagonals via Kronecker products. While the KFAC optimizer, built on top of this curvature approximation, and its variants such as George et al. (2018) show promising results for medium-sized NNs (e.g. Osawa et al., 2023), their usefulness for large NN training is often limited by (i) memory consumption, and (ii) the use of low-precision floating-point (FP) training that renders matrix decompositions or inversions required to pre-condition the gradient numerically unstable.

Recently, Lin et al. (2023) proposed an inverse-free Kronecker-factored natural gradient descent (INGD) algorithm that replaces matrix inversion with subtraction in a matrix logarithm space. The algorithm's update is purely based on matrix multiplications and therefore numerically stable in single-precision (FP-32); however, it is unclear whether this extends to half-precision (FP-16). Furthermore, INGD has not been derived from the popular natural gradient approaches for DL. Hence, it is unclear if and how the method is connected to the predominant KFAC optimizer. Also, INGD does not improve over KFAC's memory complexity since its Kronecker factors are dense matrices of the same size. And lastly, INGD has only been tested on convolution-based models and it is unclear whether it is useful for training modern transformer-based architectures (Vaswani et al., 2017).

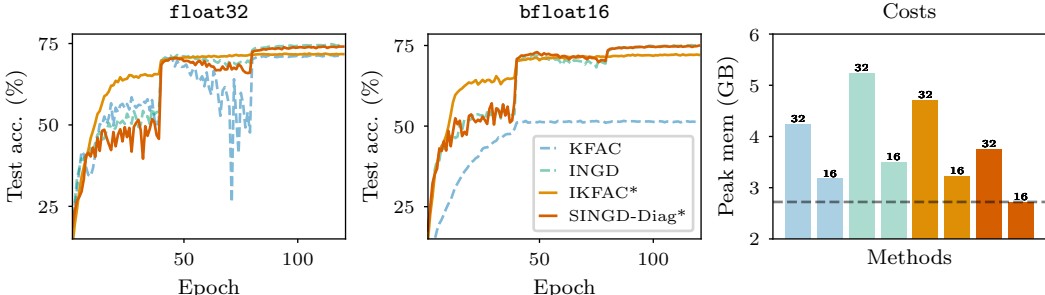

Figure 1: CIFAR-100 experiments on VGG net. *Left/Center:* Our proposed methods (IKFAC and SINGD) are stable in single *and* half precision—unlike KFAC—since they do not require matrix inversions/decompositions. IKFAC effectively performs KFAC updates and achieves similar performance in single precision. For this task, replacing the dense Kronecker factors (INGD = SINGD-Dense) with diagonal ones (SINGD-Diag) does not harm performance while reducing computations. *Right:* Memory consumption of all methods. Removing Riemannian momentum (IKFAC) or using structured Kronecker factors (SINGD-Diag) reduces INGD's memory consumption both in FP-32 and FP-16. In FP-16, SINGD-Diag achieves AdamW's memory consumption (dashed line)[1].

Here, we extend INGD to lower its computational cost and theoretically resolve its connection to other approximate NGD methods for DL (see Figure 2 for an overview): First, we show that a special case of INGD recovers the KFAC method. This allows us to effectively perform KFAC updates in an *inverse-free* fashion. We call this modification of INGD *inverse-free KFAC (IKFAC)*. Second, we exploit an algebraic structure in the matrix logarithm space and propose structure-preserving updates to maintain sparse structures on Kronecker factors. This significantly reduces the memory cost and leads to a novel, scalable second-order optimization algorithm we call *structured inverse-free natural gradient descent (SINGD)* which contains INGD and IKFAC as special cases. We evaluate SINGD on convolution- and transformer-based models and show that it can (i) be competitive with SGD and AdamW while using as little memory as the latter thanks to structured Kronecker factors and (ii) yield better performance than KFAC while being stable in half-precision. In summary:

(a) We bridge the gap between INGD (Lin et al., 2023) and the original KFAC (Martens & Grosse, 2015), which requires matrix inversions that become unstable in low precision. Thereby, we effectively make KFAC inverse-free and amenable to low-precision training (Figure 1, *left/center*).

(b) We impose structures in the Kronecker factors of the INGD update, allowing them to be sparse to lower the memory consumption (Figure 1, *right*). We analyze the impact of a range of structures on downstream task performance and find that sparse structures that considerably lower the memory consumption (even lower than AdamW) can lead to competitive performance.

(c) We show that our method, SINGD, works well on a range of modern architectures. Unlike optimizers which are useful in narrower scopes (SGD is best for convolution-based models, AdamW is best for transformers), our method can reliably train both architectures.

## 2 PRELIMINARIES

We first introduce the necessary ingredients to establish a connection between INGD and KFAC, as they are derived from different perspectives. We start by describing Newton's method as both methods can be seen as approximate Newton methods using NGD. Newton's method is a classical second-order method to solve unconstrained optimization problems. NN training often corresponds to an unconstrained minimization problem. E.g., consider training a NN for image classification. Given a set of $N$ examples $\{y_i, \mathbf{x}_i\}_{i=1}^N$ with labels $y_i$ and images $\mathbf{x}_i$, the optimization problem is

$$\min_{\boldsymbol{\mu}} \ell(\boldsymbol{\mu}; \mathbf{y}, \mathbf{X}) := \min_{\boldsymbol{\mu}} \sum_{i=1}^N c(y_i, f(\boldsymbol{\mu}; \mathbf{x}_i)), \tag{1}$$

where $\mathbf{y} := (y_1, \ldots, y_N)$ and $\mathbf{X} := (\mathbf{x}_1, \ldots, \mathbf{x}_N)$, $\hat{y}_i := f(\boldsymbol{\mu}; \mathbf{x}_i)$ is a NN that outputs a predicted label $\hat{y}_i$ for an image $\mathbf{x}_i$. Parameters $\boldsymbol{\mu}$ denote learnable weights of the NN, and $c(y_i, \hat{y}_i)$ is a

---

[1]The memory footprint of our current implementation can be reduced even further with in-place operations.

differentiable loss function to measure the difference between a true label $y_i$ and a predicted label $\hat{y}_i$. To tackle Equation (1), Newton's method follows the update

$$\boldsymbol{\mu} \leftarrow \boldsymbol{\mu} - \mathbf{S}^{-1}\left(\nabla_{\boldsymbol{\mu}}\ell(\boldsymbol{\mu}; \mathbf{y}, \mathbf{X})\right), \tag{2}$$

where $\mathbf{S} := \nabla_{\boldsymbol{\mu}}^2 \ell(\boldsymbol{\mu}; \mathbf{y}, \mathbf{x})$ is the Hessian of the loss.

## 2.1 KFAC: Approximate NGD for maximum likelihood estimation

It is often computationally intractable to compute the Hessian of a NN model as required by Newton's method. For natural gradient descent (Amari, 1998), a Fisher information matrix (FIM) is used instead of the Hessian matrix by reformulating problem (1) as maximum likelihood estimation (MLE) of a likelihood function $p(\mathbf{y}|\boldsymbol{\mu}, \mathbf{X}) = \prod_i p(y_i|\boldsymbol{\mu}, \mathbf{x}_i)$, where $p(y_i|\boldsymbol{\mu}, \mathbf{x}_i) := \exp(-c(y_i, f(\boldsymbol{\mu}, \mathbf{x}_i)))$. In this case, the minimization problem is equivalent to the MLE problem:

$$\max_{\boldsymbol{\mu}} p(\mathbf{y}|\boldsymbol{\mu}, \mathbf{X}) \iff \min_{\boldsymbol{\mu}} -\log p(\mathbf{y}|\boldsymbol{\mu}, \mathbf{X}) = \min_{\boldsymbol{\mu}} \ell(\boldsymbol{\mu}; \mathbf{y}, \mathbf{X}). \tag{3}$$

The MLE problem formulation allows us to exploit additional statistical structures such as the FIM. The FIM for the MLE problem is defined as shown below (Kunstner et al., 2019), where we assume a label $y$ is sampled from the distribution $p(y|\boldsymbol{\mu}, \mathbf{x}_i)$ given an image $\mathbf{x}_i$:

$$F(\boldsymbol{\mu}) := \sum_{i=1}^{N} \mathbb{E}_{y \sim p(y|\mu, x_i)}\left[\nabla_{\boldsymbol{\mu}} \log p(y|\boldsymbol{\mu}, \mathbf{x}_i)\nabla_{\boldsymbol{\mu}}^{\top} \log p(y|\boldsymbol{\mu}, \mathbf{x}_i)\right] = \sum_{i=1}^{N} \mathbb{E}_{y \sim p(y|\mu, x_i)}\left[-\nabla_{\boldsymbol{\mu}}^2 \log p(y|\boldsymbol{\mu}, \mathbf{x}_i)\right]. \tag{4}$$

For ubiquitous loss functions like the mean-squared error and cross-entropy, and more generally, many members of the exponential family with natural parameterization, the FIM coincides with the generalized Gauss-Newton (GGN) matrix (Wang, 2010; Martens, 2014), a common approximation of the Hessian in deep learning (Schraudolph, 2002; Botev et al., 2017). This relationship connects NGD to Newton's method. A common approximation of the FIM/GGN and Hessian is the so-called *empirical* Fisher $\hat{F}(\boldsymbol{\mu})$, which replaces the samples $y$ from the model's predictive distribution in Eq. (4) with the empirical labels $y_i$ from the data:

$$\hat{F}(\boldsymbol{\mu}) := \sum_{i=1}^{N} \nabla_{\boldsymbol{\mu}} \log p(y_i|\boldsymbol{\mu}, \mathbf{x}_i)\nabla_{\boldsymbol{\mu}}^{\top} \log p(y_i|\boldsymbol{\mu}, \mathbf{x}_i) \approx -\sum_{i=1}^{N} \nabla_{\boldsymbol{\mu}}^2 \log p(y_i|\boldsymbol{\mu}, \mathbf{x}_i) = \mathbf{S}.$$

While there is no clear theoretical justification for this approximation of the Hessian (Kunstner et al., 2019), it simplifies the implementation and reduces the cost, and has been shown to work well in practice (Graves, 2011; Osawa et al., 2019). This approximation is also known as Fisher's scoring with observed FIM for nonlinear models (Osborne, 1992; Smyth, 1996; 2015). With this, we can formulate an NGD update with the *empirical* FIM $\hat{F}(\boldsymbol{\mu})$ to approximate Newton's method as

$$\boldsymbol{\mu} \leftarrow \boldsymbol{\mu} - \beta\left(\hat{F}(\boldsymbol{\mu})\right)^{-1}\nabla_{\boldsymbol{\mu}}\ell(\boldsymbol{\mu}; \mathbf{y}, \mathbf{X}) \approx \boldsymbol{\mu} - \beta\mathbf{S}^{-1}\nabla_{\boldsymbol{\mu}}\ell(\boldsymbol{\mu}; \mathbf{y}, \mathbf{X}).$$

We call this update NGD for MLE.

Kronecker-Factored Approximate Curvature (Heskes, 2000; Martens & Grosse, 2015, KFAC) is the probably most commonly used second-order optimizer in deep learning. The KFAC algorithm is based on a Kronecker-factored approximation of the Fisher, which is also sometimes referred to as KFAC. Here, we refer to the algorithm as KFAC or KFAC method and to the approximation as Kronecker approximation. In this work, we consider the Kronecker approximation of the *empirical* Fisher. The Kronecker approximation of the Fisher approximates the FIM with a block-diagonal matrix, where each block $\tilde{F}_l$ is a Kronecker-factored matrix and corresponds to the one layer $l$ of the neural network. This approximation has first been derived for linear layers, later for convolutional (Grosse & Martens, 2016) and recurrent layers (Martens et al., 2018), and has recently been generalized to all linear layers that use weight sharing (Eschenhagen et al., 2023), e.g. also graph neural networks and transformers. The blocks are defined as $\tilde{F}_l(\boldsymbol{\mu}) := \mathbf{U}_l \otimes \mathbf{G}_l$, where $\mathbf{U}_l := \mathbf{u}_l\mathbf{u}_l^{\top} \in \mathbb{R}^{d_i \times d_i}$ and $\mathbf{G}_l := \mathbf{g}_l\mathbf{g}_l^{\top} \in \mathbb{R}^{d_o \times d_o}$. The vector $\mathbf{u}_l \in \mathbb{R}^{d_i}$ is the input to the $l$th layer and $\mathbf{g}_l \in \mathbb{R}^{d_o}$ is the gradient of the loss $\ell$ w.r.t. the output of the $l$th layer. We have suppressed the dependence on the parameters $\boldsymbol{\mu}$ and the input $\mathbf{x}_i$ and assume that there is no weight sharing for simplicity. In the KFAC method, exponential moving averages over $\mathbf{U}$ and $\mathbf{G}$ and damping are used, see Fig. 3.

While the Kronecker approximation allows for much more efficient preconditioning of the gradient, the dense Kronecker factors $\mathbf{S}_K$ and $\mathbf{S}_C$ still have to be stored and inverted at every preconditioning iteration when using the KFAC method. While the computational overhead can usually be amortized by only sparsely updating the preconditioner, this can lead to (i) numerical instability, especially in

low-precision settings, and (ii) memory issues for large models. Since low-precision training of large models is becoming the norm in fields like natural language processing, these issues will become more apparent in modern deep learning. There are multiple numerical concerns when using the KFAC method or its variants such as George et al. (2018) in the low precision setting. In PyTorch (Paszke et al., 2019) and Jax (Bradbury et al., 2018) implementations, all tensors have to be transformed into 32 bit floats since there is no support for 16 bit matrix inverse or decomposition operations. Moreover, over- or underflow can be an issue when calculating $\mathbf{G}_l$ and the gradient $\mathbf{g}_l$ has to be rescaled to improve stability. Memory constraints have previously been addressed by using a diagonal or block-diagonal approximation to the Kronecker factors (Zhang et al., 2018; Grosse et al., 2023). However, it is unclear if downstream performance can be maintained by using these simple structures.

## 2.2 INGD: Approximate NGD for Bayesian parameter estimation

Derived from Bayesian principles, the INGD method (Lin et al., 2023) directly approximates the inverse of the Hessian. We first introduce the Bayesian learning rule (BLR, Khan & Lin, 2017; Zhang et al., 2018; Khan et al., 2018; Osawa et al., 2019; Lin et al., 2020; Khan & Rue, 2021) and an inverse-free second-order method (Lin et al., 2021) as the INGD method builds on these works.

By the BLR, Newton's method to solve the MLE problem in (3) can be interpreted as another natural-gradient update to solve a variational inference problem (5) with a delta approximation (Khan & Rue, 2021). This interpretation allows us to view a precision matrix in the variational problem as Hessian estimation in the MLE problem. Thus, Lin et al. (2021) suggest reparameterizing the Hessian as the precision of the Gaussian posterior in a matrix logarithm space and exploiting the parameterization invariance of the natural-gradient update to obtain an inverse-free update scheme.

We now describe the learning rule and relate the natural gradient update to Newton's method. By the rule, we consider a Bayesian problem formulation, where neural network weights are considered random variables. We denote these weights by new parameters $\mathbf{w}$ since random variables are no longer learnable. We use a variational Gaussian distribution to approximate the posterior distribution of the random variables. Later on, we will show that the natural-gradient update of the Gaussian distribution recovers Newton's method for the learnable parameters. The mean and the precision matrix of the Gaussian will be treated as the learnable weights $\boldsymbol{\mu}$ and the Hessian estimation $\mathbf{S}$ in Newton's step (see (2)), respectively. The variational inference problem considered in the learning rule is defined as

$$\min_{\boldsymbol{\tau}} -\mathcal{L}(\boldsymbol{\tau}) := \mathbb{E}_{w \sim q(w;\tau)} \left[ -\log p(\mathbf{w}) - \log p(\mathbf{y}|\mathbf{w}, \mathbf{X}) \right] - H_q(\boldsymbol{\tau}), \qquad (5)$$

where $\mathcal{L}(\boldsymbol{\tau})$ is known as the evidence lower bound (ELBO), $\boldsymbol{\tau} = \{\boldsymbol{\mu}, \mathbf{S}\}$ are the learnable parameters of the variational distribution $q(\mathbf{w}|\boldsymbol{\tau}) = \mathcal{N}(\mathbf{w}|\boldsymbol{\mu}, \mathbf{S})$ which is a Gaussian distribution with mean $\boldsymbol{\mu}$ and precision $\mathbf{S}$. The likelihood $p(\mathbf{y}|\mathbf{w}, \mathbf{X}) = \exp(-\ell(\mathbf{w}; \mathbf{y}, \mathbf{X}))$ takes the same form considered in the MLE setting while the prior $p(\mathbf{w}) \propto \exp(-R(\mathbf{w}))$ is defined by a regularizer $R(\mathbf{w}) \geq 0$. To recover the MLE problem, we consider an uninformative prior $p(\mathbf{w})$ (i.e., $R(\mathbf{w}) = 0$). Finally, the function $H_q(\boldsymbol{\tau}) := \mathbb{E}_{w \sim q} \left[ -\log q \right]$ is the entropy of distribution $q(\mathbf{w}; \boldsymbol{\tau})$.

Similar to the MLE case, the Bayesian problem formulation also allows us to exploit additional statistical structures such as another FIM. The FIM used in the BLR is the FIM of the variational Gaussian distribution defined as

$$F(\boldsymbol{\tau}) := \mathbb{E}_{w \sim q(w|\tau)} \left[ \nabla_{\tau} \log q(\mathbf{w}|\boldsymbol{\tau}) \nabla_{\tau}^{\top} \log q(\mathbf{w}|\boldsymbol{\tau}) \right],$$

which has a closed-form expression and should not be confused with the FIM used for MLE (4).

Under the BLR, we perform NGD updates not only on $\boldsymbol{\mu}$ but also on $\mathbf{S}$. A NGD step (Khan & Rue, 2021) with the *exact* FIM $F(\boldsymbol{\tau})$ and stepsize $\beta > 0$ to update $\boldsymbol{\tau} = \{\boldsymbol{\mu}, \mathbf{S}\}$ is defined as

$$\boldsymbol{\tau} \leftarrow \boldsymbol{\tau} - \beta \left( F(\boldsymbol{\tau}) \right)^{-1} \nabla_{\tau} \left( -\mathcal{L}(\boldsymbol{\tau}) \right).$$

This is the NGD update for BLR, vis-à-vis for MLE. We can simplify this NGD update as

$$\mathbf{S} \leftarrow (1 - \beta)\mathbf{S} + \beta \mathbb{E}_{w \sim q(w;\mu,S)} \left[ \nabla_w^2 \ell(\mathbf{w}; \mathbf{y}, \mathbf{X}) \right], \quad \boldsymbol{\mu} \leftarrow \boldsymbol{\mu} - \beta \mathbf{S}^{-1} \mathbb{E}_{w \sim q(w;\mu,S)} \left[ \nabla_w \ell(\mathbf{w}; \mathbf{y}, \mathbf{X}) \right].$$

Then, to recover Newton's method in (2), we approximate the update rule above with

$$\mathbf{S} \leftarrow (1 - \beta)\mathbf{S} + \beta \nabla_{\mu}^2 \ell(\boldsymbol{\mu}; \mathbf{y}, \mathbf{X}), \quad \boldsymbol{\mu} \leftarrow \boldsymbol{\mu} - \beta \mathbf{S}^{-1} \nabla_{\mu} \ell(\boldsymbol{\mu}; \mathbf{y}, \mathbf{X}),$$

by (i) using a delta approximation at mean $\boldsymbol{\mu}$ to approximate expectations highlighted in red and (ii) setting the stepsize $\beta$ to 1.

Lin et al. (2021) suggest reparameterizing the precision matrix $\mathbf{S}$ in a matrix logarithm space and performing natural-gradient updates in this space. By performing NGD in the logarithm space, we can transform matrix inversion into matrix subtraction. We then go back directly to the space of the matrix inverse without explicitly inverting any matrix by using a truncated matrix exponential. Thus, the method is inverse-free and Newton-like since the natural-gradient update is parameterization invariant and recovers Newton's step by rephrasing the update in terms of $\mathbf{S}$.

We now describe the inverse-free method. We first re-express the precision matrix $\mathbf{S}$ using a non-singular square matrix $\mathbf{A}$ as $\mathbf{S} = \mathbf{A}^{-T}\mathbf{A}^{-1}$ and perform a natural-gradient step using the exact FIM in a tangent space (denoted by $\mathbf{M}$) of $\mathbf{A}_t$ at iteration $t$. We then construct a new map as $\mathbf{A} := \phi(\mathbf{A}_t, \mathbf{M}) := \mathbf{A}_t \mathrm{Expm}(1/2\mathbf{M})$ using both the current point $\mathbf{A}_t$ and parameter $\mathbf{M}$ as input, where $\mathrm{Expm}(\mathbf{N}) = \mathbf{I} + \sum_{j=1}^{\infty} \mathbf{N}^j/j!$ is the matrix exponential. Observe that $\mathbf{M}$ stays in a matrix logarithm space. At each iteration $k$, we use a new matrix logarithm space associated to $\mathbf{A}_k$ and generate a new origin $\mathbf{M}_0 = \mathbf{0}$ in this space to represent $\mathbf{A}_k$ since $\mathbf{A}_k \equiv \phi(\mathbf{A}_k, \mathbf{M}_0) = \mathbf{A}_k \mathrm{Expm}(\frac{1}{2}\mathbf{M}_0)$. The map $\phi$ is a *local reparameterization* map as it takes not only $\mathbf{M}$ but also $\mathbf{A}_k$ as input. Thanks to this map, the exact Fisher matrix is orthonormalized (Lin et al., 2023) at origin $\mathbf{M}_0$. Thus, a natural-gradient step becomes a (Euclidean) gradient step, which makes it easy to add Riemannian momentum into $\mathbf{A}$ (Lin et al., 2023). Importantly, this map allows us to perform updates in the space of $\mathbf{M}$ as the matrix logarithm space to avoid performing matrix inversions:

$$\mathbf{M} \leftarrow \mathbf{M}_0 - \beta\mathbf{N}, \quad \boldsymbol{\mu} \leftarrow \boldsymbol{\mu} - \beta\mathbf{A}_{t+1}\mathbf{A}_{t+1}^{\top}\nabla_{\mu}\ell(\boldsymbol{\mu}; \mathbf{y}, \mathbf{X}), \tag{6}$$

where $\mathbf{N} := \mathbf{A}_t^{\top}\nabla_{\mu}^2\ell(\boldsymbol{\mu}; \mathbf{y}, \mathbf{X})\mathbf{A}_t - \mathbf{I}$ and $\mathbf{A}_{t+1} := \phi(\mathbf{A}_t, \mathbf{M}) = \mathbf{A}_t\mathrm{Expm}(1/2\mathbf{M})$.

The above update is a Newton-like update without matrix inverse. To see that, we can reexpress the update of $\mathbf{A}$ in terms of $\mathbf{S}$:

$$\mathbf{S}_{t+1} = \mathbf{A}_{t+1}^{-T}\mathbf{A}_{t+1}^{-1} = \mathbf{A}_t^{-T}\mathrm{Expm}(\beta\mathbf{N})\mathbf{A}_t^{-1} = (1-\beta)\mathbf{S}_t + \beta\nabla_{\mu}^2\ell(\boldsymbol{\mu}; \mathbf{y}, \mathbf{X}) + O(\beta^2), \tag{7}$$

using the properties of the matrix exponential function.

Our work is built on the INGD method (summarized in Fig. 4) where $\mathbf{A} = \mathbf{K} \otimes \mathbf{C}$ is factorized by two Kronecker factors. Lin et al. (2023) suggest performing NGD on tangent spaces of the factors instead. Riemannian momentum is further introduced in the update of $\mathbf{K}$ and $\mathbf{C}$. The authors suggest using the Kronecker approximation discussed in Section 2.1 to approximate the Hessian $\nabla_{\mu}^2\ell(\boldsymbol{\mu}; \mathbf{y}, \mathbf{X})$ and truncating the matrix exponential to obtain a purely matrix-multiplication based update scheme. However, it is unclear how the proposed update is related to the KFAC update (summarized in Fig. 3) where another Kronecker factorization such as $\mathbf{S} = \mathbf{S}_K \otimes \mathbf{S}_C$ is used. Moreover, the INGD method remains memory inefficient for large neural networks due to the use of dense Kronecker factors. Last but not least, the authors only consider and evaluate the update on convolution-based models using single floating-point precision. It remains unclear whether the proposed update can be applicable in modern settings such as training transformer-based models and using half-precision for training.

## 3 OUR CONTRIBUTION: STRUCTURED INVERSE-FREE NGD

Inspired by the INGD method, we propose an inverse-free KFAC update scheme as a specific setting of the INGD method to address the numerical instability of the KFAC method for low-precision training. We show that this scheme effectively recovers the KFAC method. We then address the memory inefficiency of the KFAC and the INGD method for training large NNs such as transformer-based models by extending the INGD method.

### 3.1 INVERSE-FREE KFAC UPDATES FOR IMPROVING NUMERICAL STABILITY

We first propose a new inverse-free update scheme to mimic the behavior of the KFAC update. We refer to this update as IKFAC or the IKFAC update. We then show that IKFAC corresponds to a specific setting of the INGD method. Thus, our approach bridges the gap between INGD and KFAC, and sheds light on the difference between both methods. Our approach also makes it easy to switch from the INGD update to the KFAC update while keeping updates numerically stable.

---

[2]Pseudo-code reflects the public implementation but slightly differs from the description in Lin et al. (2023).

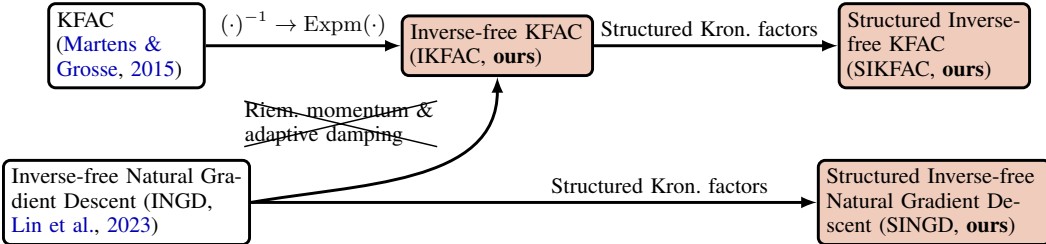

Figure 2: Overview of existing methods and their relation to our proposed methods. IKFAC behaves like KFAC (Theorem 1), but is numerically stable in lower precision. In contrast to IKFAC, INGD has Riemannian momenta, and adaptive damping and curvature which could lead to better performance in practice (Section 4). INGD is equivalent to SINGD with unstructured Kronecker factors (SINGD-Dense). Structured Kronecker factors reduce memory consumption and computational cost.

**KFAC (Martens & Grosse, 2015)**

1: Each $T$ iters, update $\mathbf{s}_K, \mathbf{s}_C$
   Obtain $\mathbf{U} \otimes \mathbf{G}$ to approximate $\nabla_\mu^2 \ell(\boldsymbol{\mu})$
   $\mathbf{S}_K \leftarrow (1 - \beta_1)\mathbf{S}_K + \beta_1 \mathbf{U}$
   $\mathbf{S}_C \leftarrow (1 - \beta_1)\mathbf{S}_C + \beta_1 \mathbf{G}$
   $\mathbf{S}_K^{-1} \leftarrow (\mathbf{S}_K + \lambda \mathbf{I}_{d_i})^{-1}$
   $\mathbf{S}_C^{-1} \leftarrow (\mathbf{S}_C + \lambda \mathbf{I}_{d_o})^{-1}$
2: $\mathbf{m}_\mu \leftarrow \alpha_2 \mathbf{m}_\mu + \mathbf{S}_C^{-1} \text{vec}^{-1}(\mathbf{g})\mathbf{S}_K^{-1} + \gamma \text{vec}^{-1}(\boldsymbol{\mu})$
3: $\boldsymbol{\mu} \leftarrow \boldsymbol{\mu} - \beta_2 \text{vec}(\mathbf{m}_\mu)$

**IKFAC (ours)**

1: Each $T$ iters, update $\mathbf{m}_K, \mathbf{m}_C, \mathbf{K}, \mathbf{C}$
   Obtain $\mathbf{U} \otimes \mathbf{G}$ to approximate $\nabla_\mu^2 \ell(\boldsymbol{\mu})$
   $\mathbf{m}_K \leftarrow 0\mathbf{m}_K + \frac{1}{2d_o}(d_o \mathbf{H}_K + \lambda d_o \mathbf{K}^\top \mathbf{K} - d_o \mathbf{I}_{d_i})$
   $\mathbf{m}_C \leftarrow 0\mathbf{m}_C + \frac{1}{2d_i}(d_i \mathbf{H}_C + \lambda d_i \mathbf{C}^\top \mathbf{C} - d_i \mathbf{I}_{d_o})$
   $\mathbf{K} \leftarrow \mathbf{K}(\mathbf{I}_{d_i} - \beta_1 \mathbf{m}_K)$
   $\mathbf{C} \leftarrow \mathbf{C}(\mathbf{I}_{d_o} - \beta_1 \mathbf{m}_C)$
2: $\mathbf{m}_\mu \leftarrow \alpha_2 \mathbf{m}_\mu + \mathbf{CC}^T \text{vec}^{-1}(\mathbf{g})\mathbf{KK}^T + \gamma \text{vec}^{-1}(\boldsymbol{\mu})$
3: $\boldsymbol{\mu} \leftarrow \boldsymbol{\mu} - \beta_2 \text{vec}(\mathbf{m}_\mu)$

Figure 3: Comparison between KFAC and IKFAC update for one weight matrix $\text{vec}^{-1}(\boldsymbol{\mu}) \in \mathbb{R}^{d_o \times d_i}$. The flattened gradient is $\mathbf{g} := \nabla_\mu \ell(\boldsymbol{\mu}) \in \mathbb{R}^{d_o d_i}$ and $\text{vec}^{-1}(\mathbf{g}) \in \mathbb{R}^{d_o \times d_i}$ is its matrix reshape. IKFAC uses $\mathbf{H}_K := \mathbf{K}^\top \mathbf{U} \mathbf{K}$ and $\mathbf{H}_C := \mathbf{C}^\top \mathbf{G} \mathbf{C}$ to incorporate the Kronecker curvature $\mathbf{U}$ and $\mathbf{G}$. Both methods use momentum buffers $\mathbf{m}_\mu$ for the weight-decayed update direction with momentum $\alpha_2$ and weight decay $\gamma$, and a learning rate $\beta_2$ for the parameter update. (*Left*) KFAC uses an exponentially moving average with decay $1 - \beta_1$ to accumulate the Kronecker factors and applies a damping term $\lambda \mathbf{I}$ before inversion to handle potential singularities in $\mathbf{S}_K, \mathbf{S}_C$. (*Right*) In contrast to KFAC, IKFAC directly approximates $(\mathbf{S}_K + \lambda \mathbf{I})^{-1}$ and $(\mathbf{S}_C + \lambda \mathbf{I})^{-1}$ by $\mathbf{KK}^\top$ and $\mathbf{CC}^\top$. The pre-conditioner update is a modification of INGD (Lin et al., 2023) and the changes—zero Riemannian momentum, and non-adaptive damping and curvature—are highlighted in red.

**INGD (Lin et al., 2023)[2]**

1: Each $T$ iterations, update $\mathbf{m}_K, \mathbf{m}_C, \mathbf{K}, \mathbf{C}$
   Obtain $\mathbf{U} \otimes \mathbf{G}$ to approximate $\nabla_\mu^2 \ell(\boldsymbol{\mu})$
   $\mathbf{m}_K \leftarrow \alpha_1 \mathbf{m}_K + \frac{1}{2d_o}(\text{Tr}(\mathbf{H}_C)\mathbf{H}_K + c^2 \mathbf{K}^\top \mathbf{K} - d_o \mathbf{I}_{d_i})$
   $\mathbf{m}_C \leftarrow \alpha_1 \mathbf{m}_C + \frac{1}{2d_i}(\text{Tr}(\mathbf{H}_K)\mathbf{H}_C + \kappa^2 \mathbf{C}^\top \mathbf{C} - d_i \mathbf{I}_{d_o})$
   $\mathbf{K} \leftarrow \mathbf{K}(\mathbf{I}_{d_i} - \beta_1 \mathbf{m}_K)$
   $\mathbf{C} \leftarrow \mathbf{C}(\mathbf{I}_{d_o} - \beta_1 \mathbf{m}_C)$
2: $\mathbf{m}_\mu \leftarrow \alpha_2 \mathbf{m}_\mu + \mathbf{CC}^T \text{vec}^{-1}(\mathbf{g})\mathbf{KK}^T + \gamma \text{vec}^{-1}(\boldsymbol{\mu})$
3: $\boldsymbol{\mu} \leftarrow \boldsymbol{\mu} - \beta_2 \text{vec}(\mathbf{m}_\mu)$

**SINGD (ours)**

1: Each $T$ iterations, update $\hat{\mathcal{L}}_{\mathbf{m}_K}, \hat{\mathcal{L}}_{\mathbf{m}_C}, \hat{\mathcal{L}}_{\mathbf{K}}, \hat{\mathcal{L}}_{\mathbf{C}}$
   Obtain $\mathbf{U} \otimes \mathbf{G}$ to approximate $\nabla_\mu^2 \ell(\boldsymbol{\mu})$
   $\hat{\mathcal{L}}_{\mathbf{m}_K} \leftarrow \alpha_1 \hat{\mathcal{L}}_{\mathbf{m}_K} + \frac{1}{2d_o}\hat{\Pi}_K(\text{Tr}(\mathbf{H}_{\hat{\mathcal{L}}_C})\mathbf{H}_{\hat{\mathcal{L}}_K} + c^2(\hat{\mathcal{L}}_{\mathbf{K}})^\top \hat{\mathcal{L}}_{\mathbf{K}} - d_o \mathbf{I}_{d_i})$
   $\hat{\mathcal{L}}_{\mathbf{m}_C} \leftarrow \alpha_1 \hat{\mathcal{L}}_{\mathbf{m}_C} + \frac{1}{2d_i}\hat{\Pi}_C(\text{Tr}(\mathbf{H}_{\hat{\mathcal{L}}_K})\mathbf{H}_{\hat{\mathcal{L}}_C} + \kappa^2(\hat{\mathcal{L}}_{\mathbf{C}})^\top \hat{\mathcal{L}}_{\mathbf{C}} - d_i \mathbf{I}_{d_o})$
   $\hat{\mathcal{L}}_{\mathbf{K}} \leftarrow \hat{\mathcal{L}}_{\mathbf{K}}(\mathbf{I}_{d_i} - \beta_1 \hat{\mathcal{L}}_{\mathbf{m}_K})$
   $\hat{\mathcal{L}}_{\mathbf{C}} \leftarrow \hat{\mathcal{L}}_{\mathbf{C}}(\mathbf{I}_{d_o} - \beta_1 \hat{\mathcal{L}}_{\mathbf{m}_C})$
2: $\mathbf{m}_\mu \leftarrow \alpha_2 \mathbf{m}_\mu + \hat{\mathcal{L}}_{\mathbf{C}}(\hat{\mathcal{L}}_{\mathbf{C}})^T \text{vec}^{-1}(\mathbf{g})\hat{\mathcal{L}}_{\mathbf{K}}(\hat{\mathcal{L}}_{\mathbf{K}})^T + \gamma \text{vec}^{-1}(\boldsymbol{\mu})$
3: $\boldsymbol{\mu} \leftarrow \boldsymbol{\mu} - \beta_2 \text{vec}(\mathbf{m}_\mu)$

Figure 4: Comparison of a single weight matrix's update between INGD and our extension—SINGD—via structured Kronecker factors. (*Left*) INGD features Riemannian momentum ($\alpha_1$), adaptive curvature ($\text{Tr}(\mathbf{H}_C)$, $\text{Tr}(\mathbf{H}_K)$), adaptive damping ($c^2 := \lambda \text{Tr}(\mathbf{C}^\top \mathbf{C})$, $\kappa^2 := \lambda \text{Tr}(\mathbf{K}^\top \mathbf{K})$), and correlated updates of $\mathbf{K}$ and $\mathbf{C}$. The pre-conditioner matrices are updated with a learning rate $\beta_1$, and the optimizer keeps a momentum buffer on the weight-decayed update with momentum $\alpha_2$ and weight decay $\gamma$. The learning rate to update the parameters is $\beta_2$. (*Right*) SINGD's update is similar but each Kronecker factor and its momentum ($\bullet$) is replaced by its structured version ($\hat{\mathcal{L}}_\bullet$, e.g. (block-)diagonal); likewise in the computation of $c^2$, $\kappa^2$, $\mathbf{H}_K$, and $\mathbf{H}_C$. When updating the momenta, their structure is preserved through a projection map $\hat{\Pi}_\bullet(\cdot)$ that restores $\hat{\mathcal{L}}_\bullet$'s structure from a dense symmetric matrix $\cdot$ (e.g. taking the (block) diagonal). Importantly, we can efficiently compute the extraction map without expanding its argument in dense form, which reduces memory and run time. The extension of IKFAC to SIKFAC is analogous.

Inspired by the INGD method, we replace matrix inversion with matrix subtraction in a matrix logarithm space and use a truncated matrix exponential map to go back to the space of the inverse matrix without explicitly inverting any matrix. The IKFAC update is related to the KFAC update as we will use $\mathbf{K}\mathbf{K}^\top$ and $\mathbf{C}\mathbf{C}^\top$ to approximate the inverse Kronecker factors $(\mathbf{S}_K + \lambda\mathbf{I})^{-1}$ and $(\mathbf{S}_C + \lambda\mathbf{I})^{-1}$ in the KFAC update, respectively. We propose the following IKFAC update with learning rate $\beta_1$ for $\mathbf{K}$ and $\mathbf{C}$ using a truncated matrix exponential[3]

$$\mathbf{K}^{\text{new}} \leftarrow \mathbf{K}\left(\mathbf{I} - \frac{\beta_1}{2}\mathbf{m}_K\right), \ \ \mathbf{C}^{\text{new}} \leftarrow \mathbf{C}\left(\mathbf{I} - \frac{\beta_1}{2}\mathbf{m}_C\right), \tag{8}$$

where $\mathbf{H}_K := \mathbf{K}^\top\mathbf{U}\mathbf{K}$, $\mathbf{H}_C := \mathbf{C}^\top\mathbf{G}\mathbf{C}$, $\mathbf{m}_K := \mathbf{H}_K + \lambda\mathbf{K}^\top\mathbf{K} - \mathbf{I}$, $\mathbf{m}_C := \mathbf{H}_C + \lambda\mathbf{C}^\top\mathbf{C} - \mathbf{I}$.

Observe that the IKFAC update in (8) is inverse-free and matrix-decomposition-free. As shown in Appendix C, $\mathbf{m}_K$ indeed stays in a matrix logarithm space since we use the truncated matrix exponential $\text{Expm}(-\frac{\beta_1}{2}\mathbf{m}_K) \approx (\mathbf{I} - \beta_1/2\,\mathbf{m}_K)$ in the update (see (8)). The logarithm space allows us to impose structural constraints on $\mathbf{K}$ as we will discuss them in the next section.

The following theorem—proof in Appendix D—formally shows that $\mathbf{K}\mathbf{K}^\top$ used in the IKFAC update is an approximation of $(\mathbf{S}_K + \lambda\mathbf{I})^{-1}$ in the KFAC update at every step even when the truncated matrix exponential is employed. Similarly, we can show $\mathbf{C}\mathbf{C}^\top$ is an approximation of $(\mathbf{S}_C + \lambda\mathbf{I})^{-1}$. Thus, the IKFAC update effectively recovers the KFAC update up to a first-order accuracy.

**Theorem 1** *If the update of $\mathbf{K}$ is updated according to the IKFAC update scheme (see Fig. 3) with the truncation of the matrix exponential and these two updates use the same initialization and the same sequence of curvature matrices $\mathbf{U}$, then the product $\mathbf{K}\mathbf{K}^\top$ has a first-order accuracy of the KFAC update of $(\mathbf{S}_K + \lambda\mathbf{I})^{-1}$ at each iteration , i.e., $\mathbf{K}\mathbf{K}^\top = (\mathbf{S}_K + \lambda\mathbf{I})^{-1} + O(\beta_1^2)$.*

Now, we show that the IKFAC scheme is a specific setting of the INGD method. As shown in Fig. 4, the INGD update of $\mathbf{K}$ without Riemannian momentum (i.e., $\alpha_1 = 0$) is

$$\mathbf{K}^{\text{new}} \leftarrow \mathbf{K}\left[\mathbf{I}_{d_i} - \frac{\beta_1}{2d_o}\big(\text{Tr}(\mathbf{H}_C)\mathbf{H}_K + \lambda\text{Tr}(\mathbf{C}^\top\mathbf{C})\mathbf{K}^\top\mathbf{K} - d_o\mathbf{I}_{d_i}\big)\right]. \tag{9}$$

Notice that $\text{Tr}(\mathbf{I}_{d_o}) = d_o$, $\mathbf{H}_C \in \mathbb{R}^{d_o \times d_o}$, $\mathbf{C} \in \mathbb{R}^{d_o \times d_o}$, and $\mathbf{K} \in \mathbb{R}^{d_1 \times d_1}$ . Thus, we can obtain IKFAC from INGD (shown in Fig. 3) by simply replacing $\text{Tr}(\mathbf{H}_C)$ and $\text{Tr}(\mathbf{C}^\top\mathbf{C})$ with $\text{Tr}(\mathbf{I}_{d_o})$:

$$\mathbf{K}^{\text{new}} \leftarrow \mathbf{K}\left[\mathbf{I}_{d_i} - \frac{\beta_1}{2d_o}\big(\text{Tr}(\mathbf{I}_{d_o})\mathbf{H}_K + \lambda\text{Tr}(\mathbf{I}_{d_o})\mathbf{K}^\top\mathbf{K} - d_o\mathbf{I}_{d_i}\big)\right]. \tag{10}$$

Note that our approach also sheds light on the difference between INGD and KFAC. In IKFAC (see Appendix C for the details), $\mathbf{H}_K$ and $\lambda\mathbf{K}^\top\mathbf{K}$ are used for incorporating curvature $\mathbf{U}$ and damping $\lambda\mathbf{I}$ in the KFAC update, respectively. In contrast, the curvature and damping is *adaptively* incorporated in INGD using $(\text{Tr}(\mathbf{H}_C)/d_o)\mathbf{H}_K$ and $(\lambda\text{Tr}(\mathbf{C}^\top\mathbf{C})/d_o)\mathbf{K}^\top\mathbf{K}$. Moreover, the updates of $\mathbf{K}$ and $\mathbf{C}$ are *correlated* in INGD due to the trace terms. In contrast, $\mathbf{K}$ and $\mathbf{C}$ are updated independently in IKFAC – just like $\mathbf{S}_K$ and $\mathbf{S}_C$ are updated independently in KFAC. These trace terms are needed to satisfy the orthonormalization condition of the Fisher matrix (Lin et al., 2023). The trace terms together with Riemannian momentum (i.e., $\alpha_1 > 0$) are missing in KFAC and IKFAC. Our experiments show that these terms could contribute to the superior performance of INGD over KFAC and IKFAC.

## 3.2 Sparse Kronecker Factors for Reducing Memory

We extend the INGD method to reduce memory consumption and lower the iteration cost for training large NNs such as transformer-based models. We propose using sparse Kronecker factors $\mathbf{K}$ and $\mathbf{C}$ in INGD. In contrast, existing structured KFAC methods (Zhang et al., 2019; Grosse et al., 2023) consider (block-)diagonal structures of factors $\mathbf{S}_K$ and $\mathbf{S}_C$. These simple structures may compromise downstream performance. Unfortunately, explicitly imposing more flexible structures on factors $\mathbf{S}_K$ and $\mathbf{S}_C$ can be either computationally challenging, memory inefficient, or numerically unstable.

Sparse factors $\mathbf{K}$ and $\mathbf{C}$ can be useful as they enable us to use more flexible structures (illustrated in Fig. 6) and achieve better downstream performance than (block-)diagonal structures. For example, a sparse factor $\mathbf{K}$ (see the leftmost plot of Fig. 7) can enforce a diagonal-plus-rank-one (dense) structure (Lin et al., 2021) in $\mathbf{S}_K$ as we use $\mathbf{K}\mathbf{K}^\top$ in the inverse-free KFAC update to approximate

---

[3]$\mathbf{K}$ and $\mathbf{C}$ could be singular if using this first-order truncation. They can be non-singular if we use a second-order truncation of the exponential. Nevertheless, the first-order truncation works very well in practice.

$\left(\mathbf{S}_K + \lambda\mathbf{I}\right)^{-1}$ in the KFAC update in Sec. 3.1. Similarly, another sparse factor $\mathbf{K}$ (see the rightmost plot of Fig. 7) can introduce a diagonal-plus-rank-one (dense) structure in the inverse of $\mathbf{S}_K$. In contrast, explicitly imposing such a structure on $\mathbf{S}_K$ or its inverse could be memory inefficient.

As a general design principle, we consider special structures preserved under elementwise matrix operations (e.g., matrix subtraction and scalar multiplication) and matrix multiplication as these operations are needed for our updates. We exploit Lie-algebraic properties in the matrix logarithm space to construct sparse structures[4] of Kronecker factors $\mathbf{K}$ and $\mathbf{C}$. We construct a structure[5] by using a subspace of the logarithm space. Concretely, we construct a new local reparameterization map for $\mathbf{K}$ at iteration $t$ such as $\mathbf{K} := \boldsymbol{\psi}(\mathbf{K}_t, \mathbf{m}_K) := \mathbf{K}_t \mathrm{Expm}(1/(2\sqrt{d_i}) \hat{\Pi}_K(\mathbf{m}_K))$, where map $\hat{\Pi}_K(\mathbf{m}_K)$ projects dense input $\mathbf{m}_K$ onto a subspace. We specify a subspace so that its sparse pattern is preserved under matrix multiplication and the elementwise matrix operations.

It can be non-trivial to design such a sparse factor while maintaining downstream performance. For example, many well-known sparse factors such as a tri-diagonal matrix do not satisfy our requirements as they are not closed under matrix multiplication. In general, it can be difficult to design the projection map so that the orthonormalization condition (Lin et al., 2023) is satisfied. One particular subspace structure satisfying these requirements is the class of upper/lower triangular matrices[6]. In the triangular case, the subspace projection $\hat{\Pi}_K$ is a weighted extraction map since projecting the logarithm space onto a subspace is like projecting a dense square matrix onto a triangular matrix space. The logarithm space arising from the dense case is an ordinary (Euclidean) matrix space because the Fisher matrix with respect to $\mathbf{K}$ at $\mathbf{m}_K = \mathbf{0}$ is locally orthonormalized. The subspace projection is a weighted map since it has to satisfy the orthonormalization condition in the subspace. Technically, we define the factor $\mathbf{A} := \mathbf{K}_t \mathrm{Expm}(\hat{\Pi}_K(\mathbf{m}_K)/(2\sqrt{d_i})) \otimes \mathbf{C}_t$, where $\mathbf{C}_t$ and $\mathbf{K}_t$ are treated as constants. Given a subspace $\Omega_K \subset \mathbb{R}^{d_i \times d_i}$ in the matrix logarithm space, the subspace projection map $\hat{\Pi}_K : \mathrm{Sym}^{d_i \times d_i} \mapsto \Omega_K$ is specified by satisfying the local orthonormalization condition (Lin et al., 2023) of the Fisher matrix with respect to $\mathbf{m}_K$: $F(\mathbf{m}_K)\big|_{m_K=\mathbf{0}} := -\mathbb{E}_{w \sim q}\left[\nabla^2_{m_K} \log q(\mathbf{w}|\boldsymbol{\mu}, \mathbf{S})\right]\big|_{m_K=\mathbf{0}} = \mathbf{I}$, where $q(\mathbf{w}|\boldsymbol{\mu}, \mathbf{S})$ is the variaitonal Gaussian with mean $\boldsymbol{\mu}$ and precision $\mathbf{S} := \mathbf{A}^{-T}\mathbf{A}^{-1}$ and $\mathrm{Sym}^{d_i \times d_i}$ denotes the set of symmetric square real matrices.

We consider several sparse structures and block extensions of the triangular matrix class as illustrated in Fig. 6. For example, the subspace projection map for a diagonal structure simply extracts diagonal entries of its input. As a non-trivial example, the subspace projection map for a lower-triangular structure extracts lower-triangular entries of its input and multiples the entries below the main diagonal by 2. Table 1 summarizes structures and their projection maps considered in this work. For an efficient implementation, we only compute and store non-zero entries of $\hat{\Pi}_K(\mathbf{m}_K)$ and $\mathbf{K}$ without explicitly forming dense matrices $\mathbf{m}_K$ and $\mathbf{K}$.

By using such a subspace and its projection map, we obtain a structured INGD update scheme (see Fig. 4). We can also obtain a structured version of IKFAC. Our approach allows us to use more expressive structures than the block-diagonal structure as illustrated in Fig. 6 and 7. These structures lower not only memory consumption (shown in Table 3) but also the iteration cost (shown in Table 2).

## 4 EXPERIMENTS

To demonstrate the robustness and memory efficiency of our method, we consider image classification tasks with transformer- and convolution-based models such as "Compact-ViT" (Hassani et al., 2021), "Swin-ViT" (Liu et al., 2021), "GC-ViT" (Hatamizadeh et al., 2023), and "Rep-ViT" (Wang et al., 2023) on datasets "CIFAR-100" and "ImageWoof-10". Note that "Rep-ViT" is a CNN model inspired by transformers while "Compact-ViT" is a data-efficient transformer using convolutional tokenization.

To be memory efficient, we consider SINGD with two sparse structures, "block-diagonal" and "hierarchical", and set the sparsity parameter $k$ (defined in Appendix. E) of each structure to be 30 so that both structures use similar memory. We also consider IKFAC, INGD, KFAC, and AdamW as our baselines. Recall that our method becomes INGD if we use a dense structure. We use mixed-precision training with BFloat16 and cosine learning rate schedules for our experiments. All methods except KFAC directly support training with BFloat16. For KFAC, we have to first transform a matrix into

---

[4]These (Lie-group) structures are preserved even under matrix inversion and matrix exponential.

[5]Each matrix (Lie-group) structure is induced by a subspace of Lie-algebra of the general linear group.

[6]This class forms a matrix associative (sub)algebra.

| Subspace of the logarithm space (Lie algebra) | Lie (sub-group) structure in $\mathbf{K}$ | Subspace projection map $\hat{\Pi}(\mathbf{m})$ |
|---|---|---|
| $\begin{bmatrix} a_{1,1} & 0 & \cdots & 0 \\ a_{2,1} & a_{2,2} & & 0 \\ \vdots & \vdots & \ddots & \vdots \\ a_{d_i,1} & a_{d_i,2} & \cdots & a_{d_i,d_i} \end{bmatrix}$ | Lower-triangular (Tril.) | $\begin{bmatrix} m_{1,1} & 0 & \cdots & 0 \\ 2m_{2,1} & m_{2,2} & & 0 \\ \vdots & \vdots & \ddots & \vdots \\ 2m_{d_i,1} & 2m_{d_i,2} & \cdots & m_{d_i,d_i} \end{bmatrix}$ |
| $\begin{bmatrix} \mathbf{A}_{11} & \mathbf{0} & \cdots & \mathbf{0} \\ \mathbf{0} & \mathbf{A}_{22} & \cdots & \mathbf{0} \\ \vdots & \vdots & \ddots & \vdots \\ \mathbf{0} & \mathbf{0} & \cdots & \mathbf{A}_{qq} \end{bmatrix}$ | (Block) Diagonal ($k$ is the block size) | $\begin{bmatrix} \mathbf{M}_{11} & \mathbf{0} & \cdots & \mathbf{0} \\ \mathbf{0} & \mathbf{M}_{22} & \cdots & \mathbf{0} \\ \vdots & \vdots & \ddots & \vdots \\ \mathbf{0} & \mathbf{0} & \cdots & \mathbf{M}_{qq} \end{bmatrix}$ |
| $\begin{bmatrix} \mathbf{A}_{11} & \mathbf{A}_{12} & \mathbf{A}_{13} \\ \mathbf{0} & \mathbf{A}_{22} & \mathbf{0} \\ \mathbf{0} & \mathbf{A}_{32} & \mathbf{A}_{33} \end{bmatrix}$, $\mathbf{A}_{22}$ is diag., $\mathbf{A}_{11} \in \mathcal{R}^{d_2 \times d_2}$, $\mathbf{A}_{33} \in \mathcal{R}^{d_3 \times d_3}$ | Hierarchical ($k := d_2 + d_3$) | $\begin{bmatrix} \mathbf{M}_{11} & 2\mathbf{M}_{12} & 2\mathbf{M}_{13} \\ \mathbf{0} & \text{Diag}(\mathbf{M}_{22}) & \mathbf{0} \\ \mathbf{0} & 2\mathbf{M}_{32} & \mathbf{M}_{33} \end{bmatrix}$ |
| $\begin{bmatrix} \mathbf{A}_{11} & \mathbf{A}_{12} \\ \mathbf{0} & \mathbf{D}_{22} \end{bmatrix}$, $\mathbf{D}_{22}$ is diag., $\mathbf{A}_{11} \in \mathcal{R}^{k \times k}$ | Rank-$k$ lower-triangular | $\begin{bmatrix} \mathbf{M}_{11} & 2\mathbf{M}_{12} \\ \mathbf{0} & \text{Diag}(\mathbf{M}_{22}) \end{bmatrix}$ |
| $\begin{bmatrix} a_0 & a_1 & a_2 & \cdots & a_{(d_i-1)} \\ 0 & a_0 & a_1 & \ddots & \vdots \\ 0 & 0 & \ddots & \ddots & a_2 \\ \vdots & \ddots & \ddots & \ddots & a_1 \\ 0 & \cdots & \ddots & 0 & a_0 \end{bmatrix}$ | Upper-triangular Toeplitz (Triu-Toepl.) | $\begin{bmatrix} b_0 & 2b_1 & 2b_2 & \cdots & 2b_{(d_i-1)} \\ 0 & b_0 & 2b_1 & \ddots & \vdots \\ 0 & 0 & \ddots & \ddots & 2b_2 \\ \vdots & \ddots & \ddots & \ddots & 2b_1 \\ 0 & \cdots & \cdots & 0 & b_0 \end{bmatrix}$ $b_j := \frac{1}{d_i - j} \sum_{k=1}^{d_i-j} m_{k,k+j}$ |

Table 1: Subspaces of the logarithm space and their projection maps $\hat{\Pi}(\mathbf{m})$, where $\mathbf{m}$ is a symmetry matrix. The hierarchical structure is constructed by replacing the diagonal matrix $\mathbf{D}_{22}$ in the rank-k lower-triangular structure with another rank-k triangular matrix $\begin{bmatrix} \mathbf{A}_{22} & \mathbf{0} \\ \mathbf{A}_{23} & \mathbf{A}_{33} \end{bmatrix}$ for a better approximation.

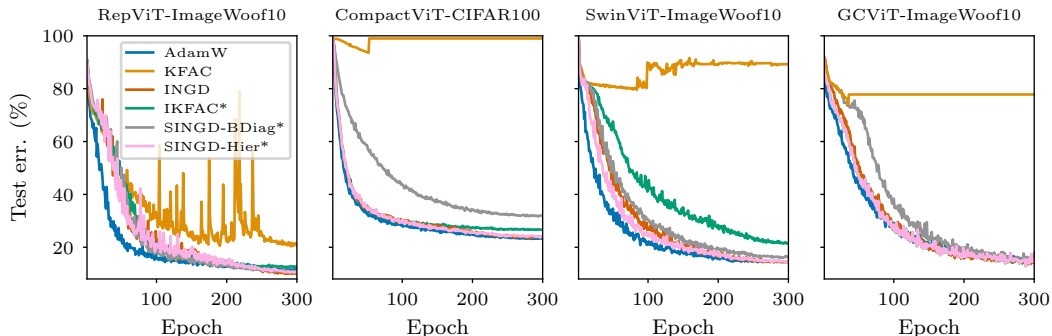

Figure 5: Test error curves for mixed-precision training in modern NN models on datasets "ImageWoof-10" and "CIFAR-100". Our method (SINGD) performs as well as INGD and Adamw while being memory efficient. SINGD and IKFAC perform stably while KFAC performs unstably.

Float32 and then transform the inverse of the matrix into BFloat16 when using matrix inversion. We fix the momentum weight to be 0.9 and tune other hyper-parameters of each optimizer using random search. The list of hyper-parameters used in the random search can be found in Table 4 in Appendix B. We use the test error to measure the performance of each method.

From Fig. 5, we can observe that SINGD with the hierarchical structure performs as well as AdamW and outperforms IKFAC. Both SINGD and IKFAC work well for mixed-precision training while KFAC performs unstably due to numerical issues. We also observe that the hierarchical structure performs as well as the dense structure (INGD) and outperforms the block-diagonal structure. Thus, we can reduce the memory consumption of INGD and make SINGD as competitive as AdamW.

## 5 CONCLUSIONS

We propose an inverse-free, memory-efficient natural gradient descent method—SINGD—which addresses the numerical instability and memory inefficiency of contemporary second-order methods like KFAC (Martens & Grosse, 2015). The algorithm is an extension of the inverse-free natural gradient (INGD) method from Lin et al. (2023), whose update relies only on matrix multiplications. We theoretically establish the algorithm's relation to KFAC by showing that a modification of INGD effectively performs KFAC-like updates. Then, we improve its memory efficiency through sparse Kronecker factors. Our experiments show that our method supports low-precision training and performs as well as AdamW for modern architecture such as transformer-based models. Our work expands the scope of second-order methods to training transformer-based NNs and training in low precision, making them more widely applicable than before. Our implementation of SINGD will be open-sourced after publication.

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

# Appendices

## A  SPACE AND TIME COMPLEXITY

| | Method | $\triangle\boldsymbol{\mu}$ (descent direction) | Update $\mathbf{S}_K$ or $\mathbf{K}$ | Update $\mathbf{S}_C$ or $\mathbf{C}$ | $\nabla_\mu\ell$ (BackProp) |
|---|---|---|---|---|---|
| Iteration Cost | KFAC | $O(d_i^2 d_o + d_o^2 d_i)$ | $O(\frac{1}{T}(md_i^2 + d_i^3))$ | $O(\frac{1}{T}(md_o^2 + d_o^3))$ | $O(md_i d_o)$ |
| | SINGD (Dense) | $O(d_i^2 d_o + d_o^2 d_i)$ | $O(\frac{1}{T}(md_i^2 + d_i^3))$ | $O(\frac{1}{T}(md_o^2 + d_o^3))$ | $O(md_i d_o)$ |
| | SINGD (Block-Diag. with block size $k$) | $O(kd_i d_o)$ | $O(\frac{1}{T}(kmd_i))$ | $O(\frac{1}{T}(kmd_o))$ | $O(md_i d_o)$ |
| | SINGD (Toeplitz) | $O(d_i d_o \log(d_o d_i))$ | $O(\frac{1}{T}(md_i \log d_i))$ | $O(\frac{1}{T}(md_o \log d_o))$ | $O(md_i d_o)$ |
| | SINGD (Rank-1 Triangular) | $O(d_i d_o)$ | $O(\frac{1}{T}(md_i))$ | $O(\frac{1}{T}(md_o))$ | $O(md_i d_o)$ |
| | SINGD (Hierarchical with parameter $k$) | $O(kd_i d_o)$ | $O(\frac{1}{T}(kmd_i))$ | $O(\frac{1}{T}(kmd_o))$ | $O(md_i d_o)$ |
| | AdamW | $O(d_i d_o)$ | NA | NA | $O(md_i d_o)$ |

Table 2: Iteration cost for a non-weight-sharing layer, where $m$ is the size of a mini-batch.

| | Method | $\nabla_\mu\ell \odot \nabla_\mu\ell$ | $\mathbf{S}_K$ or $\mathbf{K}$ | $\mathbf{S}_C$ or $\mathbf{C}$ |
|---|---|---|---|---|
| Memory Usage | KFAC | NA | $O(d_i^2)$ | $O(d_o^2)$ |
| | SINGD (Dense) | NA | $O(d_i^2)$ | $O(d_o^2)$ |
| | SINGD (Block-Diag. with block size $k$) | NA | $O(kd_i)$ | $O(kd_o)$ |
| | SINGD (Toeplitz) | NA | $O(d_i)$ | $O(d_o)$ |
| | SINGD (Rank-1 Triangular) | NA | $O(d_i)$ | $O(d_o)$ |
| | SINGD (Hierarchical with parameter $k$) | NA | $O(kd_i)$ | $O(kd_o)$ |
| | AdamW | $O(d_i d_o)$ | NA | NA |

Table 3: Additional Storage

## B  DETAILS OF THE EXPERIMENTS

| Hyperparameter | Meaning | KFAC/IKFAC/SINGD | AdamW in Fig. 4 |
|---|---|---|---|
| $\beta_2$ (AdamW: $\gamma$) | Standard stepsize | Tuned | Tuned |
| $\alpha_2$ (AdamW: $\beta_1$) | Standard momentum weight | 0.9 | 0.9 |
| $\gamma$ (AdamW: $\lambda$) | (L2) weight decay | Tuned | Tuned |
| $\lambda$ (AdmaW: $\epsilon$) | Damping | Tuned | Tuned |
| $\beta_1$ (AdamW: $1 - \beta_2$) | Stepsize for preconditioner | Tuned | Tuned |
| $\alpha_1$ | Riemannian Momentum | (SINGD only) Tuned | NA |

Table 4: Hyperparameters of all methods used for the random search

## C  CONNECTION BETWEEN IKFAC AND KFAC

To relate to the KFAC method, we now show that $\mathbf{K}^{\text{new}}\left(\mathbf{K}^{\text{new}}\right)^\top$ is an approximation of $\left(\mathbf{S}_K^{\text{new}} + \lambda\mathbf{I}\right)^{-1}$ at a new step of our scheme. For simplicity, we first assume $\mathbf{K}\mathbf{K}^\top$ exactly equals to $(\mathbf{S}_K^{\text{cur}} + \lambda\mathbf{I})^{-1}$ at the current step. Later, we will relax this assumption and prove that $\mathbf{K}\mathbf{K}^\top$ is an approximation of $(\mathbf{S}_K + \lambda\mathbf{I})^{-1}$ at every step as stated in Theorem 1. For notation simplicity, we denote $\bar{\mathbf{S}}_K := \mathbf{S}_K + \lambda\mathbf{I}$. The update of $\mathbf{S}_K$ with damping $\lambda\mathbf{I}$ can be reexpressed as an update of $\bar{\mathbf{S}}_K$:

$$\left(\mathbf{S}_K^{\text{new}} + \lambda\mathbf{I}\right) = \bar{\mathbf{S}}_K^{\text{new}} \leftarrow (1 - \beta_1)\bar{\mathbf{S}}_K^{\text{cur}} + \beta_1\left(\mathbf{U} + \lambda\mathbf{I}\right).$$

Since $\hat{\mathbf{S}}_K^{\text{cur}} = \mathbf{K}^{-T}\mathbf{K}^{-1}$ by our assumption, we can express update of $\mathbf{S}_K$ in terms of $\mathbf{K}$ as follows.

$$\bar{\mathbf{S}}_K^{\text{new}} \leftarrow (1 - \beta_1)\bar{\mathbf{S}}_K^{\text{cur}} + \beta_1\left(\mathbf{U} + \lambda\mathbf{I}\right) = \mathbf{K}^{-T}\left(\mathbf{I} + \beta_1\left(\mathbf{K}^\top\mathbf{U}\mathbf{K} + \lambda\mathbf{K}^\top\mathbf{K} - \mathbf{I}\right)\right)\mathbf{K}^{-1} = \mathbf{K}^{-T}\left(\mathbf{I} + \beta_1\mathbf{m}_K\right)\mathbf{K}^{-1}$$

$\bar{\mathbf{S}}_K^{\text{new}}$ in the KFAC update can be approximated as below, where we consider $\mathbf{I} + \beta_1\mathbf{m}_K$ as an approximate of the matrix exponential $\mathrm{Expm}(\beta_1\mathbf{m}_K) \approx \mathbf{I} + \beta_1\mathbf{m}_K$ and notice that $\mathbf{m}_K$ is symmetric.

$$\bar{\mathbf{S}}_K^{\text{new}} = \mathbf{K}^{-T}\left(\mathbf{I} + \beta_1\mathbf{m}_K\right)\mathbf{K}^{-1} \approx \mathbf{K}^{-T}\mathrm{Expm}\left(\beta_1\mathbf{m}_K\right)\mathbf{K}^{-1} = \mathbf{K}^{-T}\mathrm{Expm}\left(\frac{\beta_1}{2}\mathbf{m}_K\right)^\top\mathrm{Expm}\left(\frac{\beta_1}{2}\mathbf{m}_K\right)\mathbf{K}^{-1}.$$

Informally, we can see that $\mathbf{K}^{\text{new}}\big(\mathbf{K}^{\text{new}}\big)^{\top}$ approximates $\big(\bar{\mathbf{S}}_K^{\text{new}}\big)^{-1}$ by using the matrix exponential. We can see that $\mathbf{m}_K$ stays in a matrix logarithm space.

$$\big(\bar{\mathbf{S}}_K^{\text{new}}\big)^{-1} \approx \mathbf{K}\text{Expm}\Big(-\frac{\beta_1}{2}\mathbf{m}_K\Big)\text{Expm}\Big(-\frac{\beta_1}{2}\mathbf{m}_K\Big)^{\top}\mathbf{K}^{\top} \approx \mathbf{K}\Big(\mathbf{I} - \frac{\beta_1}{2}\mathbf{m}_K\Big)\Big(\mathbf{I} - \frac{\beta_1}{2}\mathbf{m}_K\Big)^{T}\mathbf{K}^{\top} = \mathbf{K}^{\text{new}}\big(\mathbf{K}^{\text{new}}\big)^{\top}$$

Theorem 1 formally shows that $\mathbf{K}\mathbf{K}^{\top}$ used in our update is an approximation of $\big(\mathbf{S}_K + \lambda\mathbf{I}\big)^{-1}$ in the KFAC update for every step even when the truncation of the matrix exponential is employed.

## D  PROOF OF THEOREM 1

We first consider the following lemmas in order to prove Theorem 1.

Recall that we denote $\bar{\mathbf{S}}_K := \mathbf{S}_K + \lambda\mathbf{I}$. For notation simplicity, we will drop the subscript $K$ in this section and use $\bar{\mathbf{S}}_t$ to denote $\bar{\mathbf{S}}_K$ at iteration $t$. Notice that $\bar{\mathbf{S}}_t$ is non-singular at each iteration $t$ so that we can inverse it in the original KFAC update (see Fig. 3).

**Lemma 1**  *Consider the following update in the original KFAC update at iteration $t$.*

$$\bar{\mathbf{S}}_t := (1 - \beta_1)\bar{\mathbf{S}}_{t-1} + \beta_1\big(\hat{\mathbf{U}}_{t-1} + \lambda\mathbf{I}\big)$$

*where $\mathbf{S}_t$ is the factor $\mathbf{S}_K$ used in the original KFAC update, $\beta_1$ is known as the weight of the moving average, and $\hat{\mathbf{U}}_{t-1}$ is a curvature matrix.*

*The initial factor $\bar{\mathbf{S}}_0$ can be decomposed as $\bar{\mathbf{S}}_0 = \hat{\mathbf{K}}_0^{-T}\hat{\mathbf{K}}_0^{-1}$ since $\bar{\mathbf{S}}_0$ as a preconditioning factor is symmetric positive definite.*

*Define $\hat{\mathbf{N}}_i := \hat{\mathbf{K}}_0^{T}\hat{\mathbf{U}}_i\hat{\mathbf{K}}_0 + \lambda\hat{\mathbf{K}}_0^{T}\hat{\mathbf{K}}_0 - \mathbf{I}$.*

*The Kronecker factor can be reexpressed as*

$$\bar{\mathbf{S}}_t = \hat{\mathbf{K}}_0^{-T}\left(\mathbf{I} + \beta_1\sum_{i=0}^{t-1}\hat{\mathbf{N}}_i\right)\hat{\mathbf{K}}_0^{-1} + O(\beta_1^2)$$

**Lemma 2**  *Consider the following update in our inverse-free KFAC at iteration $t$.*

$$\mathbf{K}_t := \mathbf{K}_{t-1}\left(\mathbf{I} - \frac{\beta_1}{2}\left(\mathbf{K}_{t-1}^{\top}\mathbf{U}_{t-1}\mathbf{K}_{t-1} + \lambda\mathbf{K}_{t-1}^{\top}\mathbf{K}_{t-1} - \mathbf{I}\right)\right)$$

*where $\mathbf{K}_{t-1}^{\top}\mathbf{U}_{t-1}\mathbf{K}_{t-1}$ is used in our update and $\mathbf{U}_{t-1}$ is a curvature matrix.*

*Define $\mathbf{N}_i := \mathbf{K}_i^{\top}\mathbf{U}_i\mathbf{K}_i + \lambda\mathbf{K}_i^{\top}\mathbf{K}_i - \mathbf{I}$.*

*Our update of $\mathbf{K}$ can be reexpressed as*

$$\mathbf{K}_t = \mathbf{K}_0\left(\mathbf{I} - \frac{\beta_1}{2}\sum_{i=0}^{t-1}\mathbf{N}_i\right) + O(\beta_1^2)$$

*Moreover, the product $\mathbf{K}\mathbf{K}^{\top}$ can be reexpressed as*

$$\mathbf{K}_t\mathbf{K}_t^{\top} = \mathbf{K}_0\left(\mathbf{I} - \beta_1\sum_{i=0}^{t-1}\mathbf{N}_i\right)\mathbf{K}_0^{\top} + O(\beta_1^2)$$

Lemma 3 is useful to establish a relationship between the KFAC update and our inverse-free update.

**Lemma 3**  *If we use the same sequence of curvature matrices in both the original KFAC update and our update such as $\hat{\mathbf{U}}_i = \mathbf{U}_i$ for each iteration $i$ and $\hat{\mathbf{K}}_0 = \mathbf{K}_0$ are used on the initialization, we have the following expression.*

$$\mathbf{N}_i = \hat{\mathbf{N}}_i + O(\beta_1)$$

Similarly, we have the following result for $\mathbf{C}$.

**Theorem 2** *The product $\mathbf{C}\mathbf{C}^\top$ has a first-order accuracy of the KFAC update of $\left(\mathbf{S}_C + \lambda\mathbf{I}\right)^{-1}$ at each iteration if the update of $\mathbf{C}$ is updated according to Figure 3 with the truncation of the matrix exponential and these two updates use the same initialization and the same sequence of curvature matrices $\mathbf{G}$.*

$$\mathbf{C}\mathbf{C}^\top = \left(\mathbf{S}_C + \lambda\mathbf{I}\right)^{-1} + O(\beta_1^2)$$

### D.1 PROOF OF LEMMA 1

We prove the lemma by induction We first show the base case when $t = 1$. By definition, we have

$$\bar{\mathbf{S}}_1 = (1 - \beta_1)\bar{\mathbf{S}}_0 + \beta_1\left(\hat{\mathbf{U}}_0 + \lambda\mathbf{I}\right) \tag{11}$$

$$= (1 - \beta_1)\hat{\mathbf{K}}_0^{-T}\hat{\mathbf{K}}_0^{-1} + \beta_1\left(\hat{\mathbf{U}}_0 + \lambda\mathbf{I}\right) \tag{12}$$

$$= \hat{\mathbf{K}}_0^{-T}\Big[\mathbf{I} + \beta_1\underbrace{\left(\hat{\mathbf{K}}_0^T\hat{\mathbf{U}}_0\hat{\mathbf{K}}_0 + \lambda\hat{\mathbf{K}}_0^T\hat{\mathbf{K}}_0 - \mathbf{I}\right)}_{=\hat{\mathbf{N}}_0}\Big]\hat{\mathbf{K}}_0^{-1} \tag{13}$$

$$= \hat{\mathbf{K}}_0^{-T}\left[\mathbf{I} + \beta_1\hat{\mathbf{N}}_0\right]\hat{\mathbf{K}}_0^{-1} \tag{14}$$

Thus, the claim holds when $t = 1$.

Suppose, the claim holds when $t = n$. By the claim, we have

$$\bar{\mathbf{S}}_n = \hat{\mathbf{K}}_0^{-T}\left(\mathbf{I} + \beta_1\sum_{i=0}^{n-1}\hat{\mathbf{N}}_i\right)\hat{\mathbf{K}}_0^{-1} + O(\beta_1^2) \tag{15}$$

Now, we consider the case when $t = n + 1$. Notice that

$$(1 - \beta_1)\bar{\mathbf{S}}_n = \hat{\mathbf{K}}_0^{-T}\left(\mathbf{I} + \beta_1\sum_{i=0}^{n-1}\hat{\mathbf{N}}_i - \beta_1\mathbf{I} + O(\beta_1^2)\right)\hat{\mathbf{K}}_0^{-1} + O(\beta_1^2)$$

$$= \hat{\mathbf{K}}_0^{-T}\left(\mathbf{I} + \beta_1\sum_{i=0}^{n-1}\hat{\mathbf{N}}_i - \beta_1\mathbf{I}\right)\hat{\mathbf{K}}_0^{-1} + O(\beta_1^2)$$

By the definition of $\hat{\mathbf{S}}_{n+1}$, we have

$$\bar{\mathbf{S}}_{n+1} = (1 - \beta_1)\bar{\mathbf{S}}_n + \beta_1\left(\hat{\mathbf{U}}_n + \lambda\mathbf{I}\right) \tag{16}$$

$$= \hat{\mathbf{K}}_0^{-T}\left(\mathbf{I} + \beta_1\sum_{i=0}^{n-1}\hat{\mathbf{N}}_i\underbrace{-\beta_1\mathbf{I} + \beta_1\hat{\mathbf{K}}_0^T\hat{\mathbf{U}}_n\hat{\mathbf{K}}_0 + \beta_1\lambda\hat{\mathbf{K}}_0^T\hat{\mathbf{K}}_0}_{=\beta_1\hat{\mathbf{N}}_n}\right)\hat{\mathbf{K}}_0^{-1} + O(\beta_1^2) \tag{17}$$

$$= \hat{\mathbf{K}}_0^{-T}\left(\mathbf{I} + \beta_1\sum_{i=0}^{n}\hat{\mathbf{N}}_i\right)\hat{\mathbf{K}}_0^{-1} + O(\beta_1^2) \tag{18}$$

which is exactly the claim when $t = n + 1$.

Thus, by induction, the claim holds.

### D.2 PROOF OF LEMMA 2

We prove the lemma by induction We first show the base case when $t = 1$. By definition, we have

$$\mathbf{K}_1 = \mathbf{K}_0\Big(\mathbf{I} - \frac{\beta_1}{2}\underbrace{\left(\mathbf{K}_0^\top\mathbf{U}_0\mathbf{K}_0 + \lambda\mathbf{K}_0^\top\mathbf{K}_0 - \mathbf{I}\right)}_{=\mathbf{N}_0}\Big) \tag{19}$$

Thus, the claim holds when $t = 1$.

Suppose, the claim holds when $t = n$. By the claim, we have

$$\mathbf{K}_n = \mathbf{K}_0 \left( \mathbf{I} - \frac{\beta_1}{2} \sum_{i=0}^{n-1} \mathbf{N}_i \right) + O(\beta_1^2) \tag{20}$$

Now, we consider the case when $t = n + 1$. Notice that

$$\mathbf{K}_{n+1} = \mathbf{K}_n \Big( \mathbf{I} - \frac{\beta_1}{2} \underbrace{\left( \mathbf{K}_n^\top \mathbf{U}_n \mathbf{K}_n + \lambda \mathbf{K}_n^\top \mathbf{K}_n - \mathbf{I} \right)}_{=\mathbf{N}_n} \Big) \tag{21}$$

$$= \underbrace{\mathbf{K}_0 \left( \mathbf{I} - \frac{\beta_1}{2} \sum_{i=0}^{n-1} \mathbf{N}_i \right)}_{=\mathbf{K}_n - O(\beta_1^2)} \left( \mathbf{I} - \frac{\beta_1}{2} \mathbf{N}_n \right) + O(\beta_1^2) \tag{22}$$

$$= \mathbf{K}_0 \left( \mathbf{I} - \frac{\beta_1}{2} \sum_{i=0}^{n-1} \mathbf{N}_i - \frac{\beta_1}{2} \mathbf{N}_n + O(\beta_1^2) \right) + O(\beta_1^2) \tag{23}$$

$$= \mathbf{K}_0 \left( \mathbf{I} - \frac{\beta_1}{2} \sum_{i=0}^{n} \mathbf{N}_i \right) + O(\beta_1^2) \tag{24}$$

which is exactly the claim when $t = n + 1$.

Thus, by induction, the claim holds.

Notice that $\mathbf{N}_i$ by definition is symmetric. It is easy to see that

$$\mathbf{K}_t \mathbf{K}_t^\top = \mathbf{K}_0 \left( \mathbf{I} - \frac{\beta_1}{2} \sum_{i=0}^{t-1} \mathbf{N}_i \right) \left( \mathbf{I} - \frac{\beta_1}{2} \sum_{i=0}^{t-1} \mathbf{N}_i \right)^\top \mathbf{K}_0^\top + O(\beta_1^2) \tag{25}$$

$$= \mathbf{K}_0 \left( \mathbf{I} - \frac{\beta_1}{2} \sum_{i=0}^{t-1} \mathbf{N}_i \right) \left( \mathbf{I} - \frac{\beta_1}{2} \sum_{i=0}^{t-1} \mathbf{N}_i \right) \mathbf{K}_0^\top + O(\beta_1^2) \tag{26}$$

$$= \mathbf{K}_0 \left( \mathbf{I} - \beta_1 \sum_{i=0}^{t-1} \mathbf{N}_i \right) \mathbf{K}_0^\top + O(\beta_1^2) \tag{27}$$

Thus, the claim also holds.

### D.3  PROOF OF LEMMA 3

We first show the base case when $t = 1$. By the assumption, we have $\mathbf{K}_0 = \hat{\mathbf{K}}_0$. Similarly, we have $\mathbf{U}_0 = \hat{\mathbf{U}}_0$ by the assumption.

By definition, we have

$$\mathbf{N}_0 = \mathbf{K}_0^\top \mathbf{U}_0 \mathbf{K}_0 + \lambda \mathbf{K}_0^\top \mathbf{K}_0 - \mathbf{I} \tag{28}$$

$$= \hat{\mathbf{K}}_0^\top \hat{\mathbf{U}}_0 \hat{\mathbf{K}}_0 + \lambda \hat{\mathbf{K}}_0^\top \hat{\mathbf{K}}_0 - \mathbf{I} \tag{29}$$

$$= \hat{\mathbf{N}}_0 \tag{30}$$

Thus, the claim holds when $t = 0$.

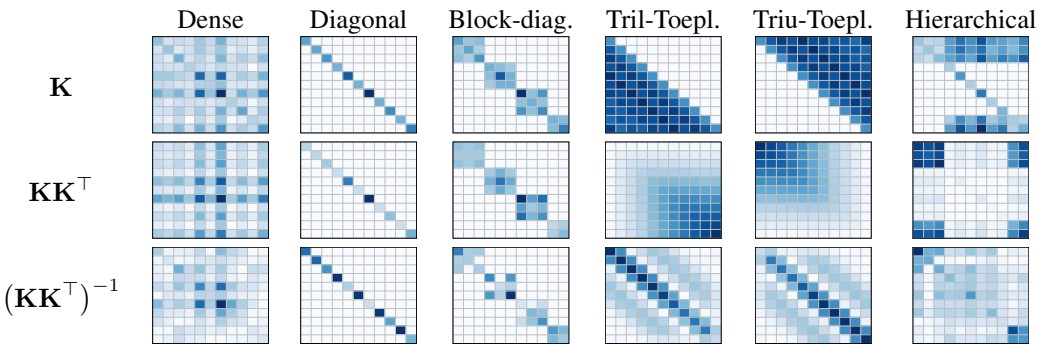

Figure 6: Illustration of structured matrices (Kronecker factors) supported by SINGD, their self-outer product (approximate inverse Hessian factor), and its inverse (approximate Hessian factor).

When $t > 0$, we can use Lemma 2 to obtain the claim. Notice that

$$\mathbf{N}_{n+1} = \mathbf{K}_{n+1}^\top \mathbf{U}_{n+1} \mathbf{K}_{n+1} + \lambda \mathbf{K}_{n+1}^\top \mathbf{K}_{n+1} - \mathbf{I} \tag{31}$$

$$= \left( \mathbf{I} - \frac{\beta_1}{2} \sum_{i=0}^n \mathbf{N}_i \right)^\top \mathbf{K}_0^\top (\mathbf{U}_{n+1} + \lambda\mathbf{I}) \mathbf{K}_0 \left( \mathbf{I} - \frac{\beta_1}{2} \sum_{i=0}^n \mathbf{N}_i \right) - \mathbf{I} + O(\beta_1^2) \text{ ( Lemma 2)} \tag{32}$$

$$= \mathbf{K}_0^\top (\mathbf{U}_{n+1} + \lambda\mathbf{I}) \mathbf{K}_0 + O(\beta_1) + O(\beta_1^2) \tag{33}$$

$$= \hat{\mathbf{K}}_0^\top (\hat{\mathbf{U}}_{n+1} + \lambda\mathbf{I}) \hat{\mathbf{K}}_0 + O(\beta_1) \tag{34}$$

$$= \hat{\mathbf{N}}_n + O(\beta_1) \tag{35}$$

### D.4 PROOF OF THEOREM 1

It is sufficient to show that the following claim holds at iteration $t$ since $\bar{\mathbf{S}}_t$ is non-singular.

$$\mathbf{K}_t \mathbf{K}_t^\top \bar{\mathbf{S}}_t = \mathbf{I} + O(\beta_1^2)$$

where we use $\bar{\mathbf{S}}_t$ to denote $\bar{\mathbf{S}}_K$ at iteration $t$.

By assumptions, we know that Lemmas 1, 2, 3 hold. Moreover, we have $\mathbf{K}_0 = \hat{\mathbf{K}}_0$. Thus, we have

$$\mathbf{K}_t \mathbf{K}_t^\top \bar{\mathbf{S}}_t = \mathbf{K}_0 \left( \mathbf{I} - \beta_1 \sum_{i=0}^{t-1} \mathbf{N}_i \right) \mathbf{K}_0^\top \bar{\mathbf{S}}_t + O(\beta_1^2) \text{ (by Lemma 2)} \tag{36}$$

$$= \mathbf{K}_0 \left( \mathbf{I} - \beta_1 \sum_{i=0}^{t-1} \mathbf{N}_i \right) \mathbf{K}_0^\top \hat{\mathbf{K}}_0^{-T} \left( \mathbf{I} + \beta_1 \sum_{i=0}^{t-1} \hat{\mathbf{N}}_i \right) \hat{\mathbf{K}}_0^{-1} + O(\beta_1^2) \text{ (by Lemma 1)} \tag{37}$$

$$= \hat{\mathbf{K}}_0 \left( \mathbf{I} - \beta_1 \sum_{i=0}^{t-1} \hat{\mathbf{N}}_i + O(\beta_1^2) \right) \left( \mathbf{I} + \beta_1 \sum_{i=0}^{t-1} \hat{\mathbf{N}}_i \right) \hat{\mathbf{K}}_0^{-1} + O(\beta_1^2) \text{ (by Lemma 3)} \tag{38}$$

$$= \hat{\mathbf{K}}_0 \mathbf{I} \hat{\mathbf{K}}_0^{-1} + O(\beta_1^2) \tag{39}$$

$$= \mathbf{I} + O(\beta_1^2) \tag{40}$$

## E SUMMARY OF SPARSE STRUCTURES

Fig. 6 and 7 demonstrate several sparse patterns in $\mathbf{K}$ and how they can be used for approximating the dense Kronecker factor $\mathbf{S}_K$.

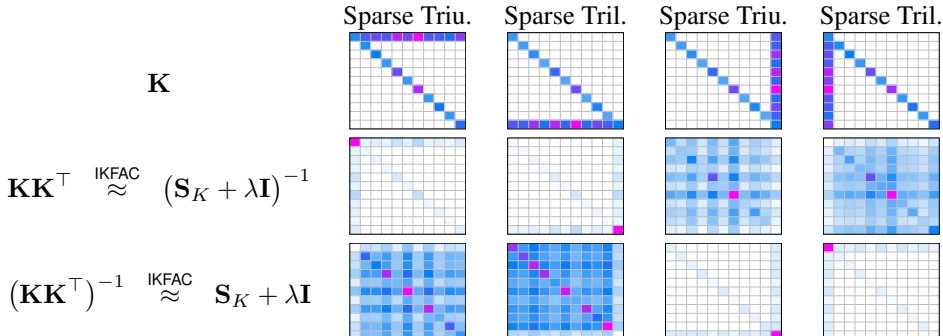

Figure 7: Imposing a dense structure on $\mathbf{K}\mathbf{K}^\top$ or its inverse using rank-one triangular matrices

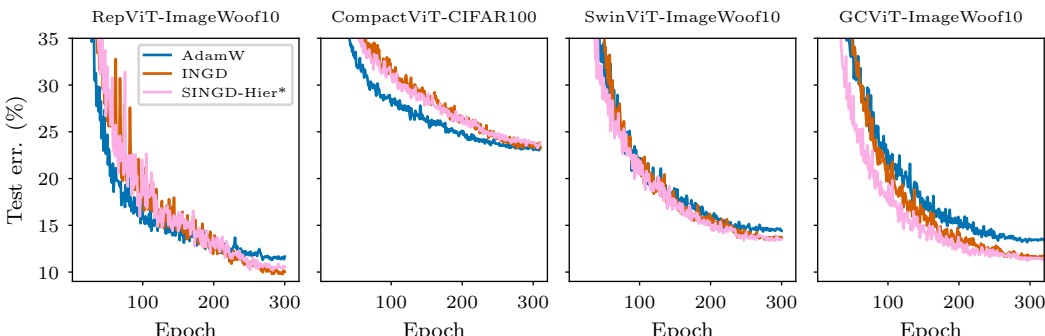

Figure 8: Test error curves for mixed-precision training in the transformer-based models with BFloat16 on datasets "CIFAR-100" and "ImageWoof-10". Our method (SINGD) performs as well as INGD while being memory efficient. SINGD and INGD can outperform AdamW in many cases.

## F  NEW RESULTS

We improve the performance of our method and INGD by updating the preconditioners more frequently in the early phase. Fig. 8 shows that our method can outperform AdamW in the models considered in Sec. 4.

We further consider convolution-based models as shown in Fig. 9 to demonstrate advantages of our method over AdamW. The convolution-based models are "GNN" (Izadi et al., 2020), "VGG", "ConvMixer" (Trockman & Kolter, 2023), and "HDVT" (Lu et al., 2022). We use a random search to find the best hyper-parameters of each of the methods. The GNN model is a graph convolution network (Kipf & Welling, 2016). We consider KFAC as a strong baseline for the GNN model. We train the GNN model with FP32 so that KFAC performs stably in this case. We also consider the diagonal structure denoted by "SINGD-Diag" since the structure may be good enough to obtain competitive performance for some models. Our approach allows users to specify a range of sparse structures of $\mathbf{K}$ and $\mathbf{C}$ in each layer.

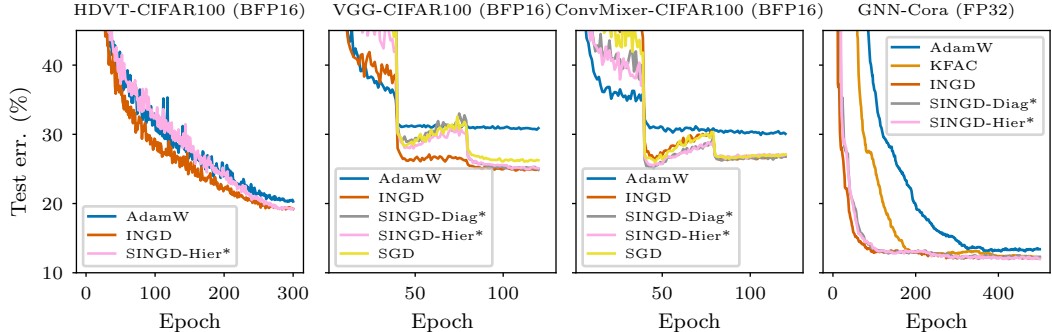

Figure 9: Test error curves for mixed-precision training in CNN and GNN models on datasets "CIFAR-100" and "Cora". Our method (SINGD) performs as well as INGD while being memory efficient. SINGD and INGD can outperform Adamw in all the models.

