# OpenReview forum: "Structured Inverse-Free Natural Gradient: Memory-Efficient & Numerically-Stable KFAC for Large Neural Nets"
_ICLR.cc/2024/Conference — Submitted to ICLR 2024_

### Official Review · Reviewer_GKer · 2023-10-29

**Soundness:** 2 fair
**Presentation:** 2 fair
**Contribution:** 2 fair
**Rating:** 5
**Confidence:** 3

**Summary:**

This paper combines the strengths of two algorithms and leverages the advantages of each to improve upon their respective limitations. Specifically, the following results and contributions are observed:

- The proposal of an inverse-free KFAC update scheme:

 Inspired by the inverse-free natural gradient descent
(INGD) algorithm, the authors introduce an inverse-free approach to the KFAC algorithm. This modification allows the KFAC algorithm to be utilized in low-precision training scenarios, eliminating the need for computationally expensive matrix inverses.
The authors propose an inverse-free KFAC update scheme, inspired by the inverse-free Kronecker-factored natural gradient descent
(INGD) algorithm. This scheme enables the KFAC algorithm to be used without requiring matrix inverses, making it suitable for low-precision training.

- Imposition of a Kronecker-factored structure on INGD:
The authors impose a Kronecker-factored structure on the INGD algorithm and propose structured inverse-free natural gradient descent (SINGD). This structural modification allows for the utilization of sparse structures, effectively reducing memory costs. By leveraging the sparsity of specific components within the algorithm, the authors aim to optimize memory utilization while maintaining computational efficiency.

**Strengths:**

Overall, this paper is well-motivated and provides a clear and detailed description of the KFAC and INGD methods. It successfully bridges the gap between these two methods by addressing their respective limitations. Specifically, the paper achieves two key objectives:
1. Making KFAC inverse-free for low-precision training: The authors propose an inverse-free KFAC update scheme, which eliminates the need for computationally expensive matrix inverses. This modification enables the KFAC algorithm to be effectively used in low-precision training scenarios, where precision is reduced to reduce memory and computational requirements.

2. Reducing memory cost in INGD: By imposing a Kronecker-factored structure on the INGD algorithm, the authors achieve a lower memory cost. This structural modification leverages sparse structures, optimizing memory utilization while maintaining computational efficiency.

**Weaknesses:**

This paper presents an incremental improvement by combining the strengths of existing algorithms and addressing their limitations. However, it fails to demonstrate completely new elements that offer significant advantages over previous approaches. Furthermore, upon reviewing the original INGD algorithm, it appears that the authors of the original work already discussed the sparse structures by considering sparse group structures in K and C. Consequently, the imposition of a Kronecker-factored structure on the INGD algorithm may be considered trivial, and not a substantial contribution compared to the original INGD algorithm. As a result, the overall strength of this work may not be sufficient for it to be accepted as a significant advancement in the field.

After rebuttal, I agree that this paper provides fair improvements on the existing algorithms KFAC and INGD.

**Questions:**

See the weakness above.

---

> ### Author Response · Authors · 2023-11-20
> **Responses to Reviewer GKer**
>
> Thanks for your feedback.
>
> > incremental improvement by combining the strengths of existing algorithms and addressing their limitations.
>
> We believe it is non-trivial to bridge the gap between INGD and KFAC. Both methods are quite different and it is left unclear by Lin et al. 2023 how they relate. One of our main contributions is to establish this previously unknown connection. Please see also the general response as we discuss the difficulty of imposing useful and sparse structures on INGD without compriosing downstream performance. Please also see general response (1).
>
> By providing this theoretical argument, we allow users to replace KFAC with IKFAC or INGD without compromising performance and introducing major overhead, while expanding its scope to modern low precision training and applications beyond optimization, such as influence functions [1] and uncertainty estimation [2].
>
> Last but not least, our work is the first work empirically validating the structured second-order updates in half precision settings for many different model classes including CNNs, GNNs, and transfomers.  Please also see general response (2).
>
>
> **References:**
>
> [1] Bae, Juhan, et al. "If Influence Functions are the Answer, Then What is the Question?." Advances in Neural Information Processing Systems 35 (2022): 17953-17967.
>
> [2] Immer, Alexander, et al. "Scalable marginal likelihood estimation for model selection in deep learning." International Conference on Machine Learning. PMLR, 2021.
>
> ---
>
>
>
> > the authors of the original work already discussed the sparse structures
>
> Lin et al. 2023 mention the possibility to construct sparse structures in the outro of their work (end of Section 4). However, to us, it seems like they left a substantial discussion and evaluation for future work. They do not provide any hints how one would realize such structures and efficiently preserve and compute them throughout the update.
>
> Our work offers such a detailed discussion and evaluation. Hence, we don't believe that our work was substantially addressed by Lin et al. 2023 and therefore represents a novel contribution. If you disagree, could you point us to the page in their paper where they provide such a discussion?

---

> > ### Comment · Reviewer_GKer · 2023-11-20
> > **Response to Authors**
> >
> > Thanks for the efforts to clarify my concerns.
> > I agree with the contributions to bridge the connection between INGD and KFAC and make fair improvements on the two algorithms.

---

### Official Review · Reviewer_zZxu · 2023-11-01

**Soundness:** 3 good
**Presentation:** 3 good
**Contribution:** 3 good
**Rating:** 6
**Confidence:** 3

**Summary:**

The authors aim to address the memory inefficiency and the numerical instability issue of KFAC. For the theory part, they close the gap between INGD and KFAC. For the experiment part, they show that the inverse-free methods have better numerical stability than KFAC. Furthermore, they show that by imposing structures on the Kronecker factors, they can achieve competitive performance while being more memory efficient.

**Strengths:**

1. The authors give a theoretical connection between an existing inverse-free method (INGD) and KFAC.

2. The authors provide a plausible reason for the better empirical performance of INGD compared to KFAC and IKFAC.

**Weaknesses:**

1. It feels like the contribution is mostly theoretical. The novelty of the inverse-free part is undermined by INGD. The discussion and analysis on the structured part also seems not to be very in-depth.
2. The empirical evaluation is a bit weak. The visualization makes it hard to see how the proposed methods compared to Adam near the end. Also, I think the evaluation will be more reasonable under settings where KFAC outperforms Adam. Such examples can be found in some previous work like [1].
3. The application of the block diagonal structure seems to be straightforward and existing. So it feels important that the proposed structure should be shown to solidly outperform block diagonal or some other simpler structure to have contribution on this part, but the discussion and the evaluation does not seem to be in-depth. The hierarchical structure seems to be interesting and outperforms the block diagonal structure, but the exact setting for the experiment is a bit unclear. For clarity, it is better to show how model weights with more than two dimensions are merged into two dimensions and the resulting dimensionality. Also, papers like [1] seem to suggest that for CNN, it is plausible for one Kronecker factor to be small and dense, while the other to be large but diagonal. This kind of structure is interesting to be considered/compared with, and it also suggests that setting the sparsity parameter to be the same for both Kronecker factors might not be the best for comparison.

[1] Bahamou, A., Goldfarb, D., & Ren, Y. (2023, April). A Mini-Block Fisher Method for Deep Neural Networks. In International Conference on Artificial Intelligence and Statistics (pp. 9191-9220). PMLR.

**Questions:**

The following is a list where a response from the authors could make my opinion of the paper more positive.

1. I feel like evaluations on settings where KFAC outperforms Adam can convince me more on the effectiveness of the proposed method.
2. More thorough discussion and comparison to show that the hierarchical structure performs better than the other simpler ones can also provide more contribution on the empirical part.
3. It is also possible for authors to justify if the theoretical contribution is enough to cover up the weakness on the empirical evaluation part.

---

> ### Author Response · Authors · 2023-11-20
> **Responses to Reviewer zZxu**
>
> Thanks for your feedback.
>
> >More thorough discussion on sparse structures such as the hierarchical structure
>
> We have updated Sec 3.2 to include more thorough discussion on sparse Kronecker factors including the hierarchical structure. Moreover, we further improve Figure 5 (now, Figure 6 in Appendix E) and readers can clearly see that the  hierarchical structure give a better approximation while keeping the computational cost low.
> Please also see the general response (1).
>
>
> > The empirical evaluation is a bit weak. The visualization makes it hard to see how the proposed methods compared to Adam near the end.  The evaluation will be more reasonable under settings where KFAC outperforms Adam.
>
> Our method can indeed outperform AdamW in many models such as CNNs, GNNs and even transformers. We include additional results in Appendix F.
> Please also see the general response (2).

---

> > ### Comment · Reviewer_zZxu · 2023-11-22
> >
> > I appreciate the authors for the added results. I am considering changing my score.
> >
> > I am curious why for three of the plots in Figure 9, SINGD uses the diagonal structure instead of the block-diagonal or the hierarchical structure. It seems important to me to have the hierarchical structure work well consistently as this is the main structural innovation. Alternatively, the authors can elaborate if the diagonal and block-diagonal structure are novel and non-trivial.

---

> ### Author Response · Authors · 2023-11-22
> **Responses to Reviewer zZxu**
>
> >   why for three of the plots in Figure 9, SINGD uses the diagonal structure instead of the block-diagonal or the hierarchical structure
>
> Note that both block-diagonal and the hierarchical structures include the diagonal structure as a special case.
> In these three models, our experiments show that the diagonal structure is good enough to obtain competitive performance.
> Thus, we can further reduce the peak memory by using the diagonal structure.
> We will add the block-diagonal and the hierarchical structures in the next revision of the paper.
> We will update the caption to clarify this point.
>
>
> In summary, we show that the hierarchical structure can achieve similar performance as the dense structure (INGD) in all the eight models. In some models, the diagonal structure is good enough to match the performance of INGD.  Our approach allows users to choose a range of sparse structures in $\mathbf{K}$ and $\mathbf{C}$ at each layer.
>
> Thank again for asking this follow-up question. Please let us know if you have any questions.

---

> > ### Author Response · Authors · 2023-11-22
> > **Responses to Reviewer zZxu**
> >
> > Dear  Reviewer zZxu,
> >
> > We updated Figure 9 in Appendix F to include the hierarchical structure in the three plots. Moreover, we **deliberately** used the same hyper-parameters for the hierarchical structure as the diagonal structure to demonstrate that the diagonal structure is a special case of the hierarchical structure.
> >
> > Please let us know if you have any follow-up questions.

---

> > > ### Comment · Reviewer_zZxu · 2023-11-22
> > >
> > > I appreciate the effort of the authors, and therefore raising my score to 6.
> > >
> > > I am still not fully convinced the contribution on the structure part is significant, and think the paper can be further improved with more endeavor on that part.

---

### Official Review · Reviewer_bReV · 2023-11-01

**Soundness:** 2 fair
**Presentation:** 3 good
**Contribution:** 3 good
**Rating:** 6
**Confidence:** 3

**Summary:**

The authors describe a novel variant of INGD, sparsifying updates in the INGD step. Structured Kronecker factors are also used to obtain a matrix-free variant of KFAC. These approaches allow for low-precision training and memory savings over their respective matrix based variants which suffer from numerical issues due to the numerical instabilities of solving lin. systems in low precision. The approach is well-written but could use some more experimental results to demonstrate the effectiveness of the method.

**Strengths:**

- well-written theory exposure that draws a bridge between INGD and KFAC
- The method is well-motivated and and well-explained.

**Weaknesses:**

- While Figure 1 provides memory footprints, these could be better-described and contextualised. Also timings would have been a welcome addition.
- It would have been great to have some more details on the hierarchical approach. Even with the SM, I could not understand it well.

**Questions:**

- How do overall training times compare to the non-matrix free variants and to SGD-based approaches such as ADAM?
- From the figures alone, it seems that these methods do no outperform ADAM in terms of convergence to a certain test-loss. What would be the advantage of using this method over ADAM?
- how exactly does the hierarchical approach work?

---

> ### Author Response · Authors · 2023-11-20
> **Responses to  Reviewer bReV**
>
> We thank the reviewer for taking the time to read and comment on this work.
>
>
> > how exactly does the hierarchical approach work?
>
> We have updated Sec 3.2 to include more thorough discussion on sparse Kronecker factors including the hierarchical structure. Moreover, we further improve Figure 5 (now Figure 6 in Appendix E) and readers can clearly see that the  hierarchical structure give a better approximation while keeping the computational cost low.
> Please also see the general response (1).
>
> ---
> > How do overall training times compare to the non-matrix free variants and to SGD-based approaches such as ADAM?
>
> Table 2 in Appendix A shows the theoretical results of training time. We will include training time for all eight models in the next version due to the limited computational budget and time constraint.
>
> ---
> > What would be the advantage of using this method over ADAM?
>
> Our method, using the right settings, can indeed outperform AdamW in many models such as CNNs, GNNs, and even transformers. We include additional results in appendix F. Please also see the general response (2).

---

> > ### Comment · Reviewer_bReV · 2023-11-22
> >
> > I thank the authors for their response. I have decided to keep the current score.

---

### Official Review · Reviewer_qqwe · 2023-11-01

**Soundness:** 3 good
**Presentation:** 3 good
**Contribution:** 3 good
**Rating:** 5
**Confidence:** 3

**Summary:**

In this paper, the authors propose a method called Structured Inverse-Free Natural Gradient Descent (SINGD) to address the memory inefficiency and numerical instability issues of second-order methods like KFAC for training large neural networks. SINGD combines an inverse-free update and imposes sparse structures of each Kronecker factor. Experimental results on transformer-based and convolution-based models demonstrate that SINGD outperforms KFAC in terms of memory efficiency and numerical robustness.

**Strengths:**

This paper is well-written and organized.  This work presents SINGD method to improving the efficiency and stability of second-order optimization methods for deep learning.  Under certain conditions, they make a connection between the INGD method and the KFAC method. Furthermore, they extend the original INGD and develop memory-efficient SINGD method by imposing special structures in the Kronecker factors.

**Weaknesses:**

This article provides a very detailed and specific introduction to KFAC and INGD method. However, it seems not highlight the contribution of this article itself. Some details on sparse kronecker factors are put in the appendix or skipped. Some derivation process and proofs of the algorithm could be included in the main text rather than in the appendix, which would make the algorithm more naturally presented.


More detailed comparisons with other state-of-the-art optimization methods such as Shampoo[1] and NGPlus[2] would strengthen the case for the superiority of SINGD. In Fig. 6, it is better to show the changes of the test error with respect to the training time. Besides, it is important to report the peak memory for these four neural networks.

[1] Anil, Rohan, et al. "Scalable second order optimization for deep learning." arXiv preprint arXiv:2002.09018 (2020).

[2] Yang, Minghan, et al. "An efficient fisher matrix approximation method for large-scale neural network optimization." IEEE Transactions on Pattern Analysis and Machine Intelligence 45.5 (2022): 5391-5403.

**Questions:**

From my experience, for experiment VGG16 on CIFAR100 dataset, the KFAC method should be tuned carefully. I want to ask if the search space of hyperparamters of KFAC method is large enough?

The difference between this paper and the INGD paper should be stated clearly and properly. As shown in Figure 4, the difference is that the use of projection $\hat{\Pi}_K$ and the structures $\hat{\mathcal{L}}$ in SINGD. Could you please give a specific example to show how to compute the projection and how to use the strcuture?

Why the NGD update for BLR can be simplified as the equations in the end of Page 4? Please add some explanation.

Does $O(\beta_1^2)$ small enough in Theorem 1?

---

> ### Author Response · Authors · 2023-11-20
> **Responses to  Reviewer qqwe**
>
> Thanks for your feedback.
>
> ---
> > However, it seems not highlight the contribution of this article itself. Some details on sparse kronecker factors are put in the appendix or skipped.
>
> In the last paragraph of the introduction, we highlight the contribution of introducing sparse structures.
> We do agree that focussing more on the sparse structures we introduce will make our contribution clearer. We have updated the manuscript according to the changes in our global response to emphasize the difficulty of constructing a structures and give concrete examples of sparse factors and their projection maps.
>
> Please also see the general response (1).
>
> ---
>
> > Comparisons with other state-of-the-art optimization
>
> One of our contributions is to enable KFAC optimizers in lower precision settings and expand the scope of KFAC methods in modern low precision training. Thus, our main focus is on enabling KFAC in lower precision settings.
> Moreover, to our best knowledge, many existing second-order optimizers such as Shampoo are numerically unstable in half precision settings.
>
> Thus, we believe that KFAC (in Float32) and AdamW (in BFloat16) are strong baselines in our experiments.
>
> Please also see the general response (2). We also consider other models such as graph NNs and convolution NNs.
>
> ---
>
> > Report Peak memory and training time
>
> Tables 2-3 in Appendix A show the theoretical results of peak memory and training time. We will include them for all eight models in the next verison due to the limited computational budget and time constraint.
>
> ---
>
> > I want to ask if the search space of hyperparamters of KFAC method is large enough?
>
> We use a random search in the log scale to find the hyperparamters of KFAC include learning rate, weight decay, damping, the moving average term $\beta_1$ to update $S_K$ and $S_C$. The search space of each hyperparameter contains multiple orders of magnitude.
>
> ---
>
> >A specific example to show how to compute the projection and how to use the structure
>
> Table 4 show four examples of computing the projection and the usage of sparse structures.
> We moved Table 4 in the appendix into the main text. This Table is Table 1 in the new revision.
>
> ---
> > Why the NGD update for BLR can be simplified as the equations in the end of Page 4? Please add some explanation.
>
> As shown in Appendix C of Khan et al. 2018 [1], the NGD update for BLR can be simplified as
> $\mu = \mu- \beta S^{-1} \nabla_\mu (-\mathcal{L}) =\mu -\beta S^{-1} \nabla_\mu \{ E_{w\sim q}[\ell(w)] - H(q) \}$ and $S = S + 2 \beta \nabla_{S^{-1}}  (-\mathcal{L}) = S + 2 \beta \nabla_{S^{-1}} \{ E_{w\sim q}[\ell(w)] - H(q) \}$.
>
> We further use gradient identities of a Gaussian distribution $q$ (see [2]) to obtain the update. These gradient identities are known as reparameterization gradients in variational inference.
>
>
> $\nabla_\mu \{ E_{w\sim q}[\ell(w)] - H(q) \} = \nabla_\mu  E_{w\sim q}[\ell(w)] = E_{w\sim q}[ \nabla_w \ell(w)]$ (Bonnet’s Theorem).
>
> $2 \nabla_{S^{-1}} \{ E_{w\sim q}[\ell(w)] - H(q) \} = 2\nabla_{S^{-1}}  E_{w\sim q}[\ell(w)] - S =  E_{w\sim q}[\nabla_w^2 \ell(w)] - S$ (Price’s Theorem).
>
> In the first step, we use the fact that the entropy of a Gaussian is $H(q)=\frac{1}{2} \log \mathrm{det}(S^{-1})$. We obtain the last step by using the gradient identities of the Gaussian.
>
> ---
> > Does the approximation small enough in Theorem 1?
>
> In all our experiments, we use values ranging from $\beta_1 = 10^{-3}$ to $10^{-2}$. Therefore, we believe that the second-order term $O(\beta_1^2)$ in Theorem 1 is small enough, and therefore KFAC should be close to IKFAC.
>
> **References:**
>
> [1] Khan, et al. "Fast and scalable bayesian deep learning by weight-perturbation in adam." International conference on machine learning. PMLR, 2018.
>
> [2] Lin et al. "Stein's Lemma for the Reparameterization Trick with Exponential Family Mixtures." arXiv preprint arXiv:1910.13398 (2019).

---

> > ### Comment · Reviewer_qqwe · 2023-11-23
> >
> > I appreciate the authors' responses. I will keep my rating.

---

### Author Response · Authors · 2023-11-20
**General Responses**

We thank the reviewers for taking the time to read and provide detailed feedback on our work. From their assessments, we identified two main criticisms, which we believe we can address in the following way:

**(1) Focus on structures (changes in Sec. 3.2 and Appendix E):**  We agree that the main text should contain a more thorough discussion on *sparse* Kronecker factors, as they are one of our main contributions. This point is intended to address the difference of SINGD and INGD as several reviewers raised this concern.


We have made the following changes in the main text to incorporate this feedback:

* In Sec 3.2, we will emphasize the difficulty of constructing these structures. It is non-trivial to design such sparse factors since they have to satisfy technical conditions and be computationally efficient. Many well-known sparse matrices such as tri-diagonal/banded matrices cannot be used as their structure is not preserved under matrix multiplication.

 * We have swapped Table 4 from Appendix E (mathematical description of matrix structures) with Figure 5 (graphical illustration of matrix structures) from the original main text to provide concrete examples of sparse factors and their projection maps. This table as Table 1 in the new revision illustrates that it is *not straightforward* to design these maps to satisfy the technical conditions. Further, we added a paragraph with details on the construction to Section 3.2.

* Figures 6 and 7 in Appendix E give graphical illustration of the sparse matrix structures.

* The above change also rigorously explains the hierarchical structure we introduce in this work.


**(2) Performance relative to AdamW (changes in Appendix F):**
Our method *can* indeed outperform AdamW in many models such as CNNs, GNNs, and even transformers.

* We identified that one issue with SINGD is its initially slow convergence due to the pre-conditioner's initialization. $\mathbf{K}$ and $\mathbf{C}$ are initialized to identity and need to warm up first. If we update the pre-conditioners more frequently (only in the early phase) this significantly improves SINGD's initial convergence. In fact, we found that it can even beat AdamW on the transformers in Section 4. Figure 8 in Appendix F demonstrates this.

* To demonstrate SINGD's potential on a different class of architecture, we consider training GNNs. We have added new experiments with GNNs in Figure 9 in Appendix F where SINGD outperforms AdamW.

* It is also known that AdamW often does not lead to state-of-the-art performance on CNNs [e.g. 1]. We have added new experiments with CNNs in Figure 9 in Appendix F which show that SINGD performs better than AdamW in this setting.

Thanks again for pointing us towards those points!

**References:**

[1] Wilson, Ashia C., et al. "The marginal value of adaptive gradient methods in machine learning." Advances in neural information processing systems 30 (2017).

---

### Author Response · Authors · 2023-11-22
**Any additional questions or concerns**

Dear reviewers,

We wonder if our responses address your concerns. We are happy to answer any follow-up questions. Please let us know before the end of the rebuttal phrase if you have any questions or concerns.

---

### Author Response · Authors · 2023-11-22
**Thanks for the constructive discussion!**

We thank the reviewers for their thoughtful feedback and the constructive discussion!

We believe that, while the current revision still has minor shortcomings, the contributions within our paper are substantial and should therefore be considered for publication (two reviewers raised their score during the discussion):

- To the best of our knowledge, **our work proposes the first second-order method that is practical for state-of-the-art training schemes** and closes a gap between first-order and second-order methods. Our SINGD
    - is stable, and works well, in `bfloat16`
    - only requires matrix multiplications (no decompositions/inversions)
    - works consistently on transformers, CNNs & GNNs
    - supports various structures that allow for large memory savings, often without harming down-stream performance.

- One of our theoretical contributions links SINGD to KFAC, one of the most popular second-order methods for deep learning. **SINGD's connection to KFAC allows it to be used as drop-in replacement**. This is valuable for practitioners that rely on KFAC, but want to further scale their application. E.g., they can easily switch to mixed-precision training, or a larger architecture.

- We demonstrated empirically on 8 models (transformers, CNNs, GNNs) that **using structures significantly reduces computation without harming performance**. We are the first to provide a detailed derivation how to realize a matrix-multiplication-only second-order update that preserves such structures and satisfies the technical conditions.

To further improve the impact of our work, **we will open-source our PyTorch implementation of SINGD. It  efficiently realizes all proposed structures and is easy to use (works like a built-in `torch.optim.Optimizer`).**

We agree with the reviewers that there are many interesting directions regarding the impact of structures. However, we believe this will strongly depend on the application and architecture. In the context of deep learning, we are not aware of works that studied what correlations need to be captured to achieve good performance. Given that our work is the first to demonstrate promising results in the context of optimization, we think **an in-depth study of structures for other relevant applications (e.g. influence functions, Laplace approximations) is beyond the scope of our paper**.

---

### Meta-Review · Area_Chair_Ew2T · 2023-12-07

**Metareview:**

The paper builds upon an inverse-free natural-gradient method (INGD) by Lin et al., 2023 to propose a new scalable second order optimizer for deep learning called SINGD. Compared to INGD, the method leverages sparsity to make it scalable. The paper claims this to be among the first scalable second-order methods that work well for modern deep learning.

The strength of the paper is the deep theoretical foundation bringing together optimization, geometry and probability from which the authors derive a practical second-order optimization method. The main weaknesses are 1) the incremental nature of the work taking into account Lin et al. 2023;  and 2) the limited experimental evaluation. For claiming that this method is practical for modern deep-learning, larger experiments (for example, full ImageNet and larger NLP data sets) are needed.  At this stage, while the results look encouraging, it remains unclear whether such strong claims can be fully justified. Moreover, as of now, the advantage over AdamW is not entirely clear from the plots.

I recommend to reject the paper. I encourage the authors to take the reviewers feedback into account, and for a resubmission consider simplifying the theoretical presentation of the algorithm and adding more large-scale experiments.

**Justification For Why Not Higher Score:**

The theoretical contributions compared to the original INGD algorithm by Lin et al. are incremental. For an empirical paper, the evaluation on the CIFAR-100 and ImageWoof-10 datasets is rather small-scale.

**Justification For Why Not Lower Score:**

N/A

---

### Decision · Program_Chairs · 2024-01-16

Reject